# Genetic determinants of host- and virus-derived insertions for hepatitis E virus replication

Michael Hermann Wißing[1], Toni Luise Meister [1,2,3,4], Maximilian Klaus Nocke[1,5], André Gömer[1], Mejrema Masovic[1], Leonard Knegendorf [6,7], Yannick Brüggemann[1], Verian Bader [8,9], Anindya Siddharta[6], Claus-Thomas Bock [10], Alexander Ploss [11], Scott P. Kenney[12], Konstanze F. Winklhofer[8,13], Patrick Behrendt[6,14,15], Heiner Wedemeyer[14,15,16], Eike Steinmann [1,17] ✉ & Daniel Todt [1,5] ✉

Hepatitis E virus (HEV) is a long-neglected RNA virus and the major causative agent of acute viral hepatitis in humans. Recent data suggest that HEV has a very heterogeneous hypervariable region (HVR), which can tolerate major genomic rearrangements. In this study, we identify insertions of previously undescribed sequence snippets in serum samples of a ribavirin treatment failure patient. These insertions increase viral replication while not affecting sensitivity towards ribavirin in a subgenomic replicon assay. All insertions contain a predicted nuclear localization sequence and alanine scanning mutagenesis of lysine residues in the HVR influences viral replication. Sequential replacement of lysine residues additionally alters intracellular localization in a fluorescence dye-coupled construct. Furthermore, distinct sequence patterns outside the HVR are identified as viral determinants that recapitulate the enhancing effect. In conclusion, patient-derived insertions can increase HEV replication and synergistically acting viral determinants in and outside the HVR are described. These results will help to understand the underlying principles of viral adaptation by viral- and host-sequence snatching during the clinical course of infection.

Hepatitis E virus (HEV) is the leading cause of acute viral hepatitis and has been recognized as a global health problem in developing and industrialized countries[1]. The estimated incidence of 20 million infections annually, leads to 3.3 million acute cases and 70,000 deaths[2]. Infections are usually self-limiting and asymptomatic or cause mild symptoms like fever, vomiting, abdominal pain and nausea in healthy individuals. However, infections in patients with preexisting liver disease frequently lead to acute liver failure and pregnant women show mortality rates of up to 30%[3,4].

The virus belongs to the species *Paslahepevirus balayani* within the family *Hepeviridae* and comprises eight different genotypes (GT),

five of which (HEV-1-4 and HEV-7) are pathogenic for humans[1]. The distribution of different genotypes is dependent on their reservoir and route of transmission. HEV-1 and HEV-2 solely infect humans, are transmitted via contaminated water, and are found in developing countries with poor sanitary conditions, where they can cause waterborne outbreaks and epidemics. In contrast, HEV-3, HEV-4, and HEV-7 have their reservoir primarily in pigs, boars, deer, dromedary camels (HEV-7), and other mammals. They are zoonotic, which - in contrast to the anthroponotic GTs -, can cause chronic infection in immunocompromised patients leading to the development of liver cirrhosis[3,5–7]. The treatment of those patients is limited to the reduction of

immunosuppressants, and the off-label use of the broad-spectrum antiviral agents ribavirin (RBV) or pegylated interferon-α[8,9].

HEV has a 7.2 kb single-stranded, positive-sense RNA genome, encoding for three open reading frames (ORFs)[10,11]. The capsid protein is encoded by ORF2 while ORF3 encodes a multifunctional protein needed for viral egress[1]. The nonstructural proteins are encoded as a polyprotein in ORF1 and contain domains for a methyltransferase (MeT), a putative papain-like cysteine protease (PCP), a hypervariable region (HVR), an RNA helicase (Hel), and an RNA-dependent RNA polymerase (RdRp)[1,10,12–14]. As the name implies the HVR shows the highest divergence in the HEV genome, contains a proline-rich region, and has been shown to be involved in host range adaptation[15] and modulating RNA replication[16]. Furthermore, in samples of HEV-infected patients, sequences of viral or host origin have been observed in the HVR[14,17,18]. This was also the case for the HEV-3 Kernow-C1-p6 strain, which is the most used laboratory strain in HEV research to date. This strain was isolated from the feces of a patient chronically infected with HEV-3 and a variant harboring an insertion of the ribosomal RPS17 RNA in its HVR was selected for after the passaging of the inoculum six times in cell culture. This variant was already present in the patient material at low frequency and showed enhanced replication in comparison to the clone Kernow-C1-p1, which resembles the inoculum and does not harbor any insertion in its HVR[19,20].

In this study, we report the discovery of patient and virus-derived HVR insertions that boost viral fitness in a reverse genetic cell culture system. We analyze the identified and previously described genomic rearrangements using state-of-the-art in vitro and in silico methods. A combination of nuclear localization signals (NLS), post-translational modifications (PTM), protein flexibility, and sequence-specific patterns inside and outside the HVR are key features for the replication-enhancing effect while not affecting RBV sensitivity. These findings provide insights into understanding the adaptation potential of circulating HEV strains and genetic determinants for viral replication. Deciphering the role of viral determinants of HEV sequence-snatching can provide novel insights into the genome plasticity of HEV.

## Results
### Identification of HEV insertions in a chronically HEV-infected patient

To study the intra-host evolution of HEV in chronically infected patients, we analyzed serum samples of a solid organ transplant recipient infected with HEV (Fig. 1a). The patient was HEV RNA positive for at least 426 days and succumbed to the disease after 653 days. The treatment with RBV for a period of ~250 days reduced the viral titers as well as the elevated γ-glutamyl transferase (γGT) levels before relapse of viral titers and liver enzymes was measured by qPCR and ELISA (Fig. 1a). Samples collected on day 321 after the initial detection of viral RNA was subjected to clonal sequence analysis of the HEV HVR. Sequencing of 200 colonies identified 181 clones (93.5%) containing HEV HVR sequences (Fig. 1b). Of those, the majority (125 clones, 69.1%) showed HVR rearrangements or insertions, while only a minority of 57 clones (31.5%) contained unaltered, shorter HVR sequences. The most

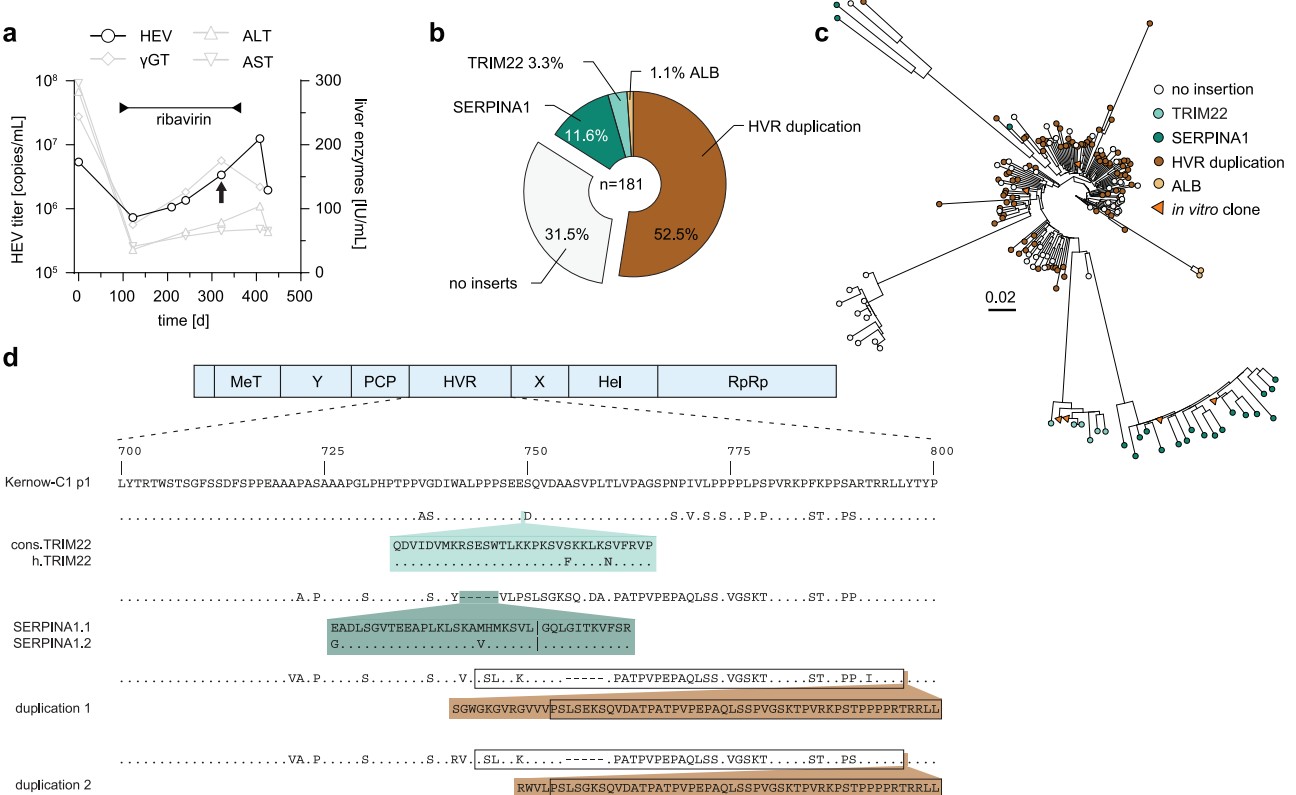

**Fig. 1 | HVR insertions in a HEV patient identified by clonal sequencing. a** The treatment course of a chronically HEV-infected solid organ recipient is shown. The solid black line indicates the viral titer as copies/mL, while the grey lines visualize the liver enzyme levels IU/mL. The treatment period with ribavirin is indicated by the bar above. The arrow highlights the time point used for clonal sequencing. **b** The distribution of HVR rearrangements identified in HEV-positive colonies is shown. Depicted is the origin of inserted sequences (genes *TRIM22, SERPINA1*, and HEV HVR duplication) and their frequency as a percentage. Source data are provided in the Source Data file. **c** Maximum likelihood tree of amplicon-based clonal sequencing derived variants. Insertion-carrying clones are depicted in cyan (*TRIM22*), green (*SERPINA1*), and brown (duplication). Sequences of engineered in vitro clones representing respective clusters are depicted as orange triangles. The scale bar indicates the average number of substitutions per site. **d** The insertion site as well as their similarity (dots) to the strain Kernow-C1-p1 are shown. Mismatches to the reference are indicated by the used amino acid as a single letter code. The inserted amino acid sequence is indicated below. The name of the constructs is indicated in front of each row. Black boxes refer to duplicated sequence snippets.

abundant type of rearrangement was duplications of HVR snippets (95 clones, 52.5%). In addition, we identified insertions of the human gene *SERPINA1* (21 clones, 11.6%), *TRIM22* (6 clones, 3.3%), and albumin (*ALB*, 2 clones, 1.1%) which have previously not been described. Sequences of all clones have been deposited at GenBank (accession numbers OR726668-OR726848). Multiple sequence alignment and phylogenetic analysis revealed considerable diversity of the HVR sequences, even within respective clusters of insertions, underlining the high genomic plasticity of this domain (Fig. 1c). Two representative consensus sequences of each cluster were genetically engineered into the Kernow-C1 p6 backbone, replacing the *RPS17* insertion and flanking regions within the HVR. Amino acid substitutions and the location of the insertions are depicted in relation to the parental strain p1 Kernow-C1 (Fig. 1d).

The sequence of a *TRIM22* insertion identified in the patient material (h.*TRIM22*) contained two non-synonymous mutations, when compared to the human consensus sequence of this gene (cons.*TRIM22*) (Fig. 1d). *TRIM22* is a E3 ubiquitin ligase and has been reported as an interferon-stimulated gene (ISG)[21]. The 20 sequences coding for the *SERPINA1* insertion, which is a gene of the serpin superfamily and also known as alpha-1 antitrypsin, differed from each other by having few point mutations and two constructs (*SERPINA1*.1 and *SERPINA1*.2) were used for follow-up analysis (Fig. 1d). The two most abundant duplications were based on HVR sequences, which differed in length, but contained a similar 3' sequence (Fig. 1d). In summary, we describe host- and viral-insertions including ISG-derived sequences in a chronically infected HEV patient.

## Enhanced replication and viral production efficiency of identified HEV insertions

To investigate the function of these insertions in the context of the HEV replication cycle, the insertions identified in the patient were introduced into a *Gaussia* luciferase replicon as well as the full-length viral genome of the Kernow-C1-p6 strain, where the insertions replaced the *RPS17* RNA insertion containing HVR. The constructs Kernow-C1-p1 and -p6 were included as references for strains with low and high replication capacity, respectively. Both *SERPINA1* insertions did not substantially abrogate replication as compared to the parental strain p6. (Fig. 2a, left panel). We observed similar replication kinetics of Kernow-C1 harboring the patient-derived *TRIM22* sequences with the point mutations. Notably, insertion of the *TRIM22* consensus sequence (cons.*TRIM22*) led to a slight reduction of luciferase signal suggesting a functional importance of the sequence polymorphism (Fig. 2a, middle panel). Although differing in length, both duplications of viral gene snippets reduced replication only ~2.5 fold in comparison to p6 96 h p.t. (post-transfection) (Fig. 2a, right panel). To rule out host gene-specific effects, we tested additional partial domains of the *TRIM22* gene (zinc finger domain and coiled-coil domain, GenBank accession number PP408296- PP408297). We observed a marked reduction compared to the constructs described above, indicative of determinants beyond the length of insertions (Supplementary. Fig. 1).

As all insertions were identified in a patient undergoing continuous RBV treatment, we used the replicon system to test whether these insertions possibly increased the resistance to RBV inhibition. The sensitivity of the identified insertions to RBV was not altered demonstrating that increased replication efficiency did not affect the RBV susceptibility (Fig. 2b). Next, we utilized the full-length reverse genetic system to investigate possible effects of the insertions on the full viral replication cycle (Fig. 2c). All generated constructs bearing insertions of *SERPINA1*.1, *SERPINA1*.2, cons.*TRIM22*, h.*TRIM22* and duplication 1 showed similar production capacity of infectious particles in comparison to p6. They reached titers of $2.0 \times 10^5$ FFU/mL (p6 level), $3.6 \times 10^5$ FFU/mL (1.8-fold increase), $1.2 \times 10^5$ FFU/mL (1.7-fold reduction), $3.1 \times 10^5$ FFU/mL (1.5-fold increase) and $1.3 \times 10^5$ FFU/mL (1.5-fold decrease), respectively. As shown before, duplication 2

replicated to equal levels as duplication 1, however, the produced titer of $5.2 \times 10^4$ FFU/mL (39-fold decrease) was much lower almost reaching p1 levels (Fig. 2c).

## Characterization of databank-derived HEV insertions

Recently, Lhomme et al. described several insertions in the HVR of acute and chronic HEV-infected patients[18,22]. The length of the inserted sequences varied from 44-71 amino acids with an average of 52 amino acids being introduced (Fig. 3a). Some of the insertions were composed of fragments of human zinc finger protein 787 (*ZNF787*), inter-alpha-trypsin inhibitor heavy chain H2 (*ITIH2*) and duplications of the HVR. Moreover, an isolate harboring sequence derived from the eukaryotic translation elongation factor 1 alpha 1 (*EEF1A1*) also encoded a stretch of amino acids of unknown origin. An additional insertion, glycine amidinotranferase (*GATM*), was identified after replication in a cell culture system[22]. To systematically evaluate whether these additional insertions in the HVR are also able to increase the viral fitness, we generated replicons and recombinant full-length genomes of the indicated insertions and assessed viral fitness and RBV susceptibility (Fig. 3a). All insertions showed productive HEV replication capacities. While insertions of kinesin family member 1B (*KIF1B*) or *ITIH2* sequence did not reach p6 levels, others such as ring finger protein 19A (*RNF19a*), *GATM* and *EEF1A1A* exhibited replication levels similar to those of p6 (Fig. 3b). Genomes harboring *RPL6* and *ZNF787* sequence even surpassed the p6 replication levels at 96 h p.t.(Fig. 3b).

The RBV titration revealed only minor changes in $IC_{50}$ values indicating no major difference in RBV sensitivity (Fig. 3c). All insertion-containing strains demonstrated a high capacity for virus production leading to absolute titers ranging from $1.3 \times 10^5$ FFU/mL (*ITIH2*, 1.6-fold reduction) up to $3.4 \times 10^5$ FFU/mL (RPL6, 1.7-fold increase) compared to $2.0 \times 10^5$ FFU/mL for p6 (Fig. 3d). In sum, all tested insertions within the HVR boosted HEV replicative fitness.

## In silico analysis of in-vivo identified insertions

Next, we hypothesized that a prerequisite for viral and host transcript snatching is a high level of expression of the original host gene or viral-dependent induction of host gene expression. Analyzing single-cell sequencing data of the human liver atlas[23] for the expression levels of the insertion encoding transcripts, we identified most transcripts in at least 25% of all hepatocytes and cholangiocytes (Fig. 4a). In addition, KIF1B and RNF19A were highly expressed by 50% of hepatocytes and at least in 25% of cholangiocytes. Some liver resident leukocytes also expressed transcripts such as *EEF1A1*, *RPL6*, *TRIM22* and *SERPINA1* (Fig. 4a). All inserted RNA transcripts were above the 0.5 reads per kilobases of transcript per 1 million mapped reads (RPKM) threshold in expression data of primary human hepatocytes (PHH) infected with HEV[24] (Fig. 4b) and are therefore defined as expressed to biologically relevant levels in PHH based on the definition of the expression atlas of the European Molecular Biology Laboratory. In detail, all transcripts were in the range of 10 to 1000 RPKM and were hence at high expression level, while some genes such as *EEF1A1* (all time points), *SERPINA1* (168 h) and ZNF787 (all time points despite 168 h) were above the 1000 RPKM threshold and are high abundantly expressed (Fig. 4b). Moreover, expression levels were not altered by productive infection of HEV over time. To test if any of the transcripts are regulated by interferon (IFN), we used transcriptome data of IFN-α treated PHH[25] and observed a significant induction of *TRIM22* confirming its ISG status in PHH (Fig. 4d). It has previously been described, that HEV is capable of downregulating IFN induced gene expression[26]. In contrast to *MxI*, *IFIT1*, *IFIT3*, and *CXCL10*, HEV does not interfere with *TRIM22* upregulation upon viral genome replication (Supplementary. Fig. 2). Subsequently, the identified and in-vitro characterized insertions in the HVR were analyzed with *in-silico* tools to identify the underlying molecular mechanism for the observed replication advantage. We predicted the protein structure of the insertion containing HVRs by

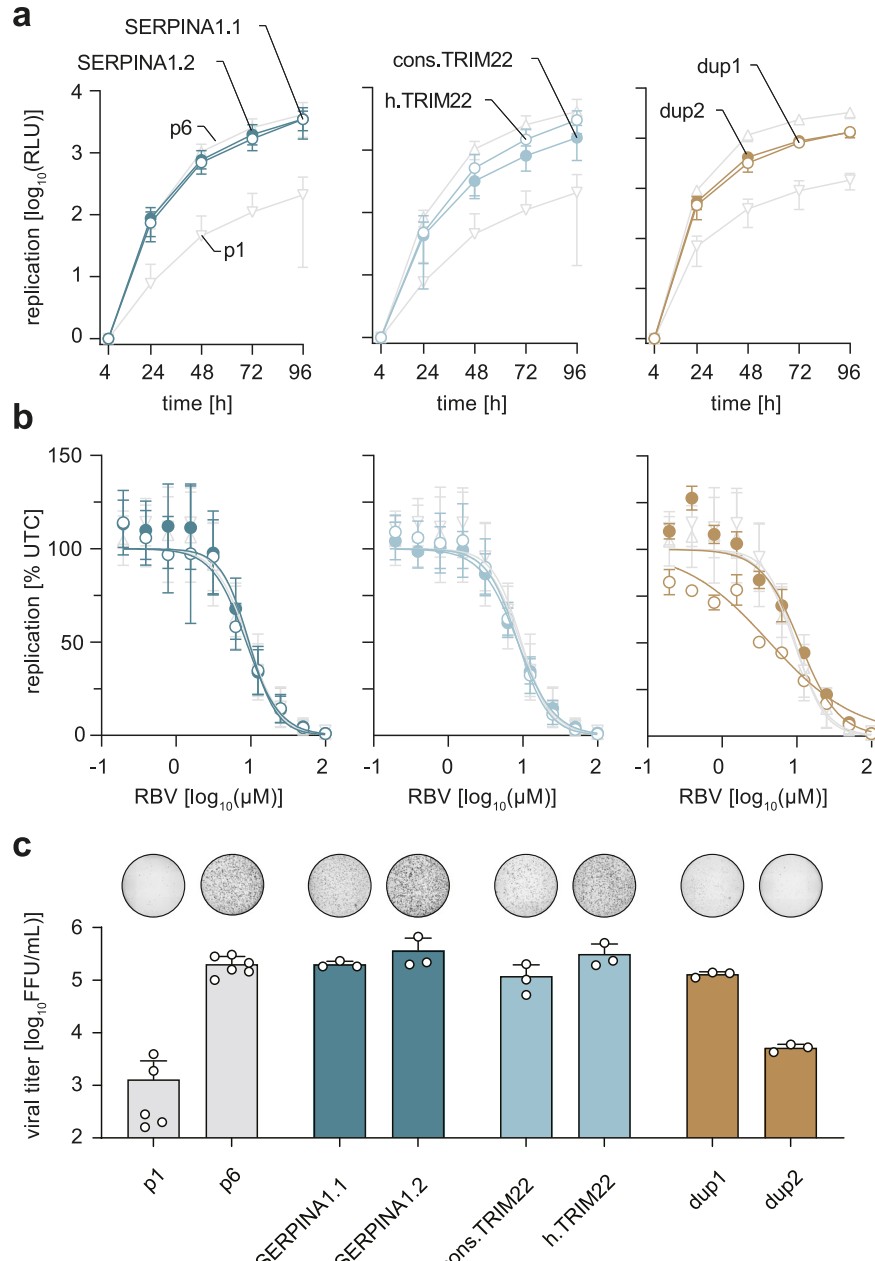

**Fig. 2 | In-vitro characterization of HVR sequences identified by clonal sequencing.** The identified insertions were cloned into the HEV reporter or full genome of the Kernow-C1-p6 strain, thereby replacing the HVR with insertion containing HVRs as indicated in Fig. 1. **a** Replication kinetics of HVR constructs with Kernow-C1-p6 (p6) and Kernow-C1-p1 (p1) as references are shown. Plotted is the time post electroporation as well as mean (+/- SD) relative light units (RLU) normalized to the four-hour value of $n = 7$ biologically independent experiments for p1 h.TRIM22, $n = 3$ for dup constructs and $n = 6$ for other constructs. **b** The HEV replicon system was used to analyse the ribavirin (RBV) sensitivity by treating the cells for five days post-electroporation with RBV concentrations ranging from 0.19 μM to 100 μM. Plotted is the mean (+/- SD) HEV replication as a percentage of untreated controls in $n = 3$ biologically independent experiments for dup constructs, $n = 6$ for other constructs. Lines represent dose-response curves of four-parameter log-logistic analysis. **c** The full-length system was used to produce infectious particles, which were titrated onto HepG2/C3A cells to determine the achieved viral titers as FFU/mL via immunofluorescence. A representative picture of a whole 96-well infected with non-enveloped HEVcc stained for the ORF2 protein (black) is shown above each column. Plotted are the means with standard deviation (+/- SD) of $n = 8$ biologically independent experiments for p1 and p6, $n = 3$ for other constructs. Source data are provided in the Source Data file.

using AlphaFold2 to analyze the impact of insertions on HVR tertiary structure. The predictions show high prediction confidence (IDDT score) for the PCP and helicase domain, which are well and consistently folded throughout all constructs (Supplementary. Fig. 3 and Supplementary. Movies 1–3). However, the HVRs presented low IDDT scores and no common structure indicating that the HVR acts as the intrinsically disordered region as proposed by Purdy et al.[15].

Next, the amino acid sequences of the HVR with incorporated insertions were analyzed for a group of post-translational

modifications (PTM) via Musite (a deep-learning framework for protein post-translational modification site prediction), for acetylation via GPS-Pail and for ubiquitination via BDM-PUB. We determined a predicted increase in the number of phosphoserines, O-linked glycosylation, phosphothreonine, acetylation and ubiquitination in comparison to p1 (Fig. 4e). The number of those predicted PTMs was increased by up to 13 (O-linked glycosylation for duplication 1) in this case on an HVR length of only 156 AA (Fig. 4e). Moreover, an increase in hydroxyproline was predicted. No obvious changes could be noted for

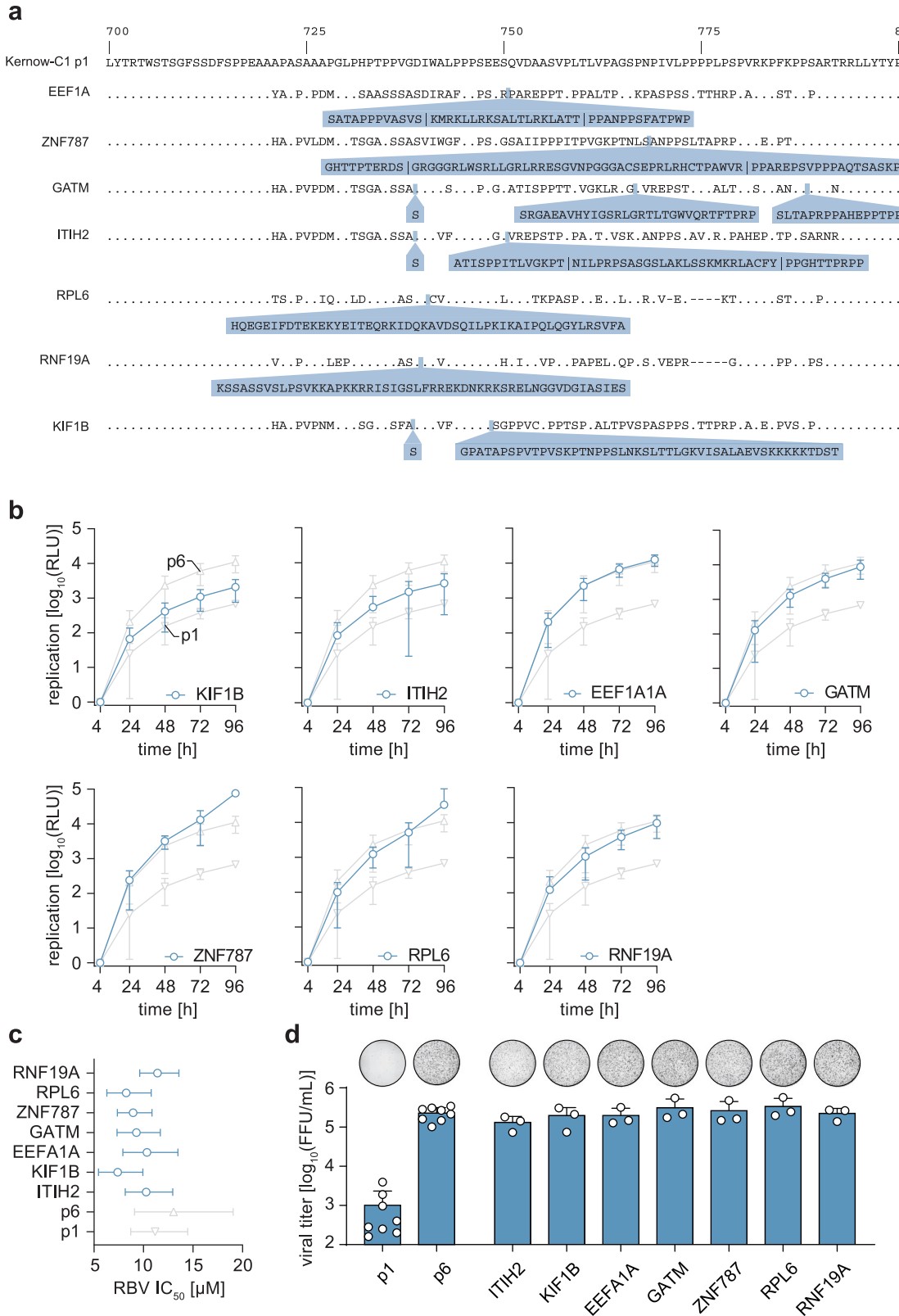

phosphotyrosine, methyllysine, N-linked glycosylation, methylargi-nine, and pyrrolidone carboxylic acid PTM frequencies (Fig. 4e). Fur-thermore, the HEV-GLUE database was used to extract full-length sequences of HEV genomes. We performed multi-sequence alignments (MSA) via Clustal Omega to identify sequences without HVR insertions ($N = 289$), which were translated and compared to the amino acid sequences of identified insertions by the tool Composition Profiler.

The analysis was restricted to the HVR with the boundaries as defined by Muñoz-Chimeno et al.[14]. The hydrophobic amino acids as defined by Eisenberg et al.[27] were significantly depleted in insertion containing HVRs ($p \leq 0.00125$), while polar amino acids, as defined by Zimmerman et al.[28] and positively charged amino acids were significantly enriched ($p \leq 0.00125$) (Fig. 4f). In line, a small enrichment of arginine and a prominent, more than 3-fold enrichment, in the usage of the amino

**Fig. 3 | In-vitro analysis of publicly available insertions. a** The insertions reported by Lhomme et al.[18] were analysed for their insertion site in relation to the strain Kernow-C1-p1. The insertion site as well as their similarity (dots) to the strain Kernow-C1-p1 are shown. Mismatches to the reference are indicated by the used amino acid as a single letter code. The inserted amino acid sequence is indicated in blue. The name of created constructs is mentioned in front of each row. **b** The replicon system was used to determine their impact on replication versus the reference strain Kernow-C1-p6 (p6) and Kernow-C1-p1 (p1). Plotted is the time post-electroporation as well as the mean (+/- SD) relative light units (RLU) normalized to the four-hour value of $n = 4$ biologically independent experiments for p1, $n = 5$ for other constructs. **c** The replicon was used to analyse the ribavirin (RBV) sensitivity by treating the cells for five days post-electroporation with RBV concentrations ranging from 0.19 μM to 100 μM. A non-linear regression and the $IC_{50}$ values were calculated using GraphPad Prism. Depicted are the respective $IC_{50}$ values and confidence intervals (CI 95%) of $n = 3$ biologically independent experiments. **d** The HEV full-length system was used to produce infectious particles, which were titrated onto HepG2/C3A cells to determine viral titers as FFU/mL via immuno-fluorescence. A representative picture of a whole 96-well infected with non-enveloped HEVcc stained for the ORF2 protein (black) is shown above each column. Plotted are the means with standard deviation (+/- SD) as well as individual data points (circle) of $n = 6$ biologically independent experiments for p1 and p6, $n = 3$ for other constructs. Source data are provided in the Source Data file.

acid lysine was observed in insertion containing HVRs ($p \leq 0.00125$). Lysine is especially unique in terms of its ability to be post-translationally modified and known to be phosphorylated, ubiquitinated, methylated, acetylated, palmitoylated, hydroxylated and involved in the formation of nuclear localization signals (NLS).

## Alanine scanning mutagenesis of the NLS sequence in the RPS17 insertion of p6

Since an NLS was reported for the *RPS17* insertion of the p6 strain[29] and enrichment of the positively charged amino acid lysine was identified for insertion containing constructs, we next analyzed if an NLS sequence could be a common feature by using the tools NLS Mapper, NLSTradamus and Prosite. We further assessed nuclear localization via NucPred (Fig. 5a). The strain p1 demonstrated a low NLS prediction score in NucPred while all other tools indicated the absence of nuclear localization signals. The tools NLS Mapper and NucPred proposed an enhanced probability for NLS sequences being present in the inserted sequences compared to the p1 HVR. Moreover, for the insertions cons.*TRIM22*, duplication1, duplication2, *EEF1A1*, *RNF19A*, and *KIF1B* at least one of the tools with binary output suggested the presence of NLS sequences or nuclear localization. Hence, we aimed to test the influence of the NLS sequence in the prototypic strain p6 by disrupting the required amino acids (Fig. 5b, highlighted in green) by alanine scanning mutagenesis in a similar approach as reported before[29]. We exchanged the lysine residues at positions 32, 44, and 45 as well as the arginine at position 33 with alanine. Moreover, the lysine at position 49 was mutated to alanine by using all possible codons for alanine. Furthermore, at position 49 a silent mutation was introduced, which together with the codon-optimized version of the RPS17 insertion (Fig. 5b and Supplementary Fig. 4a) were created to differentiate between the effects of amino acid sequence versus RNA sequence on viral fitness.

The silent mutation K49K as well as the codon-optimized insertion demonstrated no decrease in luciferase activity, while all K49A constructs exhibited a decrease in replication capacity (Fig. 5c) with no change in RBV susceptibility (Supplementary Fig. 4b). The constructs with multiple amino acids exchanges revealed a cumulative decrease in replication capacity compared to the parental p6 strain when multiple Lys/Arg were mutated (Fig. 5e). Moreover, a stronger decrease of replication capacity was observed when positions 44/45 and 49 were simultaneously mutated in comparison to positions 32/33. Notably, the exchange of all five amino acids in the construct KR32/33AA_KK44/45AA_K49A reduced the replication capacity to the level of the p1 strain (Fig. 5e) indicating a possible involvement of the NLS sequence for the fitness advantage of p6. Furthermore, it is notable that the codon optimizing the *RPS17* insertion (Supplementary Fig. 4a) or introducing the synonymous K49K mutation did not reduce RNA replication implying that the protein sequence of the insertions rather than the RNA sequence/structure was responsible for the observed phenotype. The $IC_{50}$ values in the RBV titration did not differ significantly between all constructs (Supplementary Fig. 4c). The replication efficacy at the latest time point is shown in Fig. 5d–f for comparison of all constructs. All mutations were also tested in the full-

length system to validate effects on the full viral replication cycle. The virus production capacity of the virus was not altered by the K49K silent mutation or the codon-optimized version of the RPS17 insertion (Fig. 5g). However, the achieved viral titers decreased in a cumulative manner with increasing numbers of mutated Lys/Arg.

We further analyzed if the predicted NLS sequences within the HVRs of the p6, *SERPINA1*, *TRIM22*, and duplication constructs were biologically active by performing a translocation assay as described for p6 by Kenney and Meng[29]. In brief, the HVRs of the constructs p6, *SERPINA1*, *TRIM22*, and the duplications were cloned in tandem with a triple eYFP and transfected into Huh7 cells. The eYFP is expressed in the cytosol if no NLS sequence is present. In contrast, the eYFP reporter translocates into the nucleus when the HVRs contain an active NLS sequence. In line with Kenney et al.[29], the HVR of the p6 strain resulted in an increased nuclear localization compared to p1 and the mutant k, harboring all five Lys/Arg variants (Fig. 5h, i and Supplementary Movies 4–7). In the case of the described insertions, their ability to cause nuclear eYFP translation highly varied and could only be robustly observed for the *SerpinA1*.1 and *SerpinA 1*.2 variants (Fig. 5 h, i and Supplementary Movies 8–13). We next tested if down-regulation of IFN-induced gene expression (compare Supplementary Fig. 2) during HEV replication might be due to interference of HEV proteins with the cell's nuclear core import and stalling translocation of transcript factors. To exclude possible interactions and interference of the predicted NLS with the nuclear protein import cycle, we analyzed the dynamic expression of importins and karyopherin-α as well as -ß subunits in HEV-infected PHH (Supplementary Fig. 4d). While all mRNA was expressed at moderate levels, except the karyopherin-α subunit *KPNA5*, active HEV replication did not alter expression within 168 h after infection (Supplementary Fig. 4d). However, these data cannot exclude that proteins were sequestered by HEV.

These data indicate that the NLS sequence is not the only determinant for the enhanced replication efficiency of HEV insertions.

## Construction and characterization of artificial insertions in the HEV HVR

Next, we constructed multiple artificial insertions to investigate which properties of the genome rearrangements in the HVR were responsible for the enhanced replication effect. First, we generated a construct with a rigid XP-linker to mimic the insertion spacer character with high proline content and second, we constructed a more flexible, serine- and glycine-rich linker, (Fig. 6a and Supplementary Fig. 5A). To further test if an NLS sequence is a key feature, four well-characterized NLSs with different strength (*IPMK < IP3KB < SHIP1 < SV40*[30]) were inserted in the middle of the two artificial linker-based insertion constructs possessing the same length like the *RPS17* insertion (Fig. 6a and Supplementary Fig. 5a). In addition, one insertion was designed without an NLS. All artificial insertions were cloned into the luciferase reporter system of the backbone of p1 and p6 (sequences available at GenBank accession numbers OR700721-OR700740). None of the artificial insertions reached replication levels as high as p6 (Fig. 6b). Moreover, the rigid XP linker construct showed comparable replication levels to the

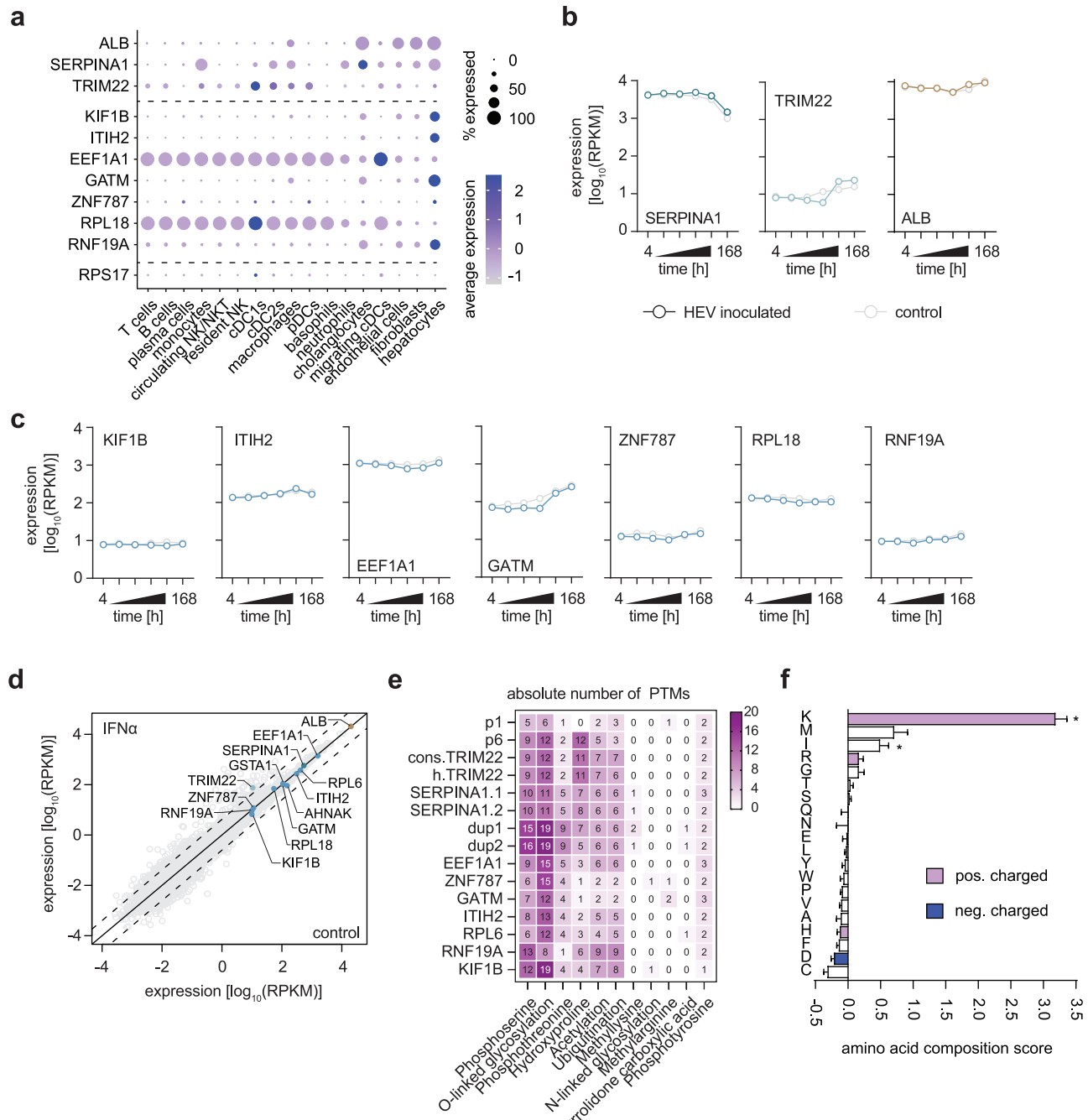

**Fig. 4 | *In-silico* analysis of HEV insertion containing HVRs. a** The liver cell atlas was used to analyse the expression values of identified insertions in various liver cell types on a single cell level. The frequency of cells expressing the gene is depicted by the size of the dot, while the colour of the dot encodes the average expression level according to Guilliams et al.[23] in those cells. **b,c** Expression of transcripts encoding the insertions was analysed in a published data set of non-infected as well as Kernow-C1-p6 infected primary human hepatocytes (PHH). Depicted is the expression as RPKM values over time for all genes. **d** Differential expression of indexed transcripts in a data set of PHH treated with PBS (control) or interferon-α (IFNα). The solid black line indicates no regulatory effect, while the dashed lines indicate 4-fold up or down-regulation in either condition. **e** HVR amino acid sequences were analysed for post-translational modification via musite, for ubiquitination via BDM-PUB, and for acetylation via GPS-Pail. The number of predicted PTM sites is plotted as a heatmap. Kernow-C1-p1 (p1) and Kernow-C1-p6 (p6)

were used as reference. **f** Amino acid composition of insertion containing HVRs (*n* = 15 sequences examined) was compared to HEV-GLUE deposited HVRs without insertions (*n* = 289 sequences examined) using the tool composition profiler. Shown are fold changes in amino acid usage of insertion containing HVRs over non-insertion containing HVRs. Positive-charged amino acids are indicated in red, while negatively charged ones are highlighted in blue. The statistical significance associated with a specific enrichment or depletion is estimated using a Bonferroni-corrected two-sample t-test between two sequences of binary indicator variables, one sequence for each of the samples (I *p*-value = 0.001988 (≤0.0025), K *p*-value = 0.0 (≤0.0025)). For the calculation of composition differences, 10,000 bootstrap iterations were used for non-parametric estimation of the confidence intervals for the reported amino acid compositions. Source data are provided in the Source Data file.

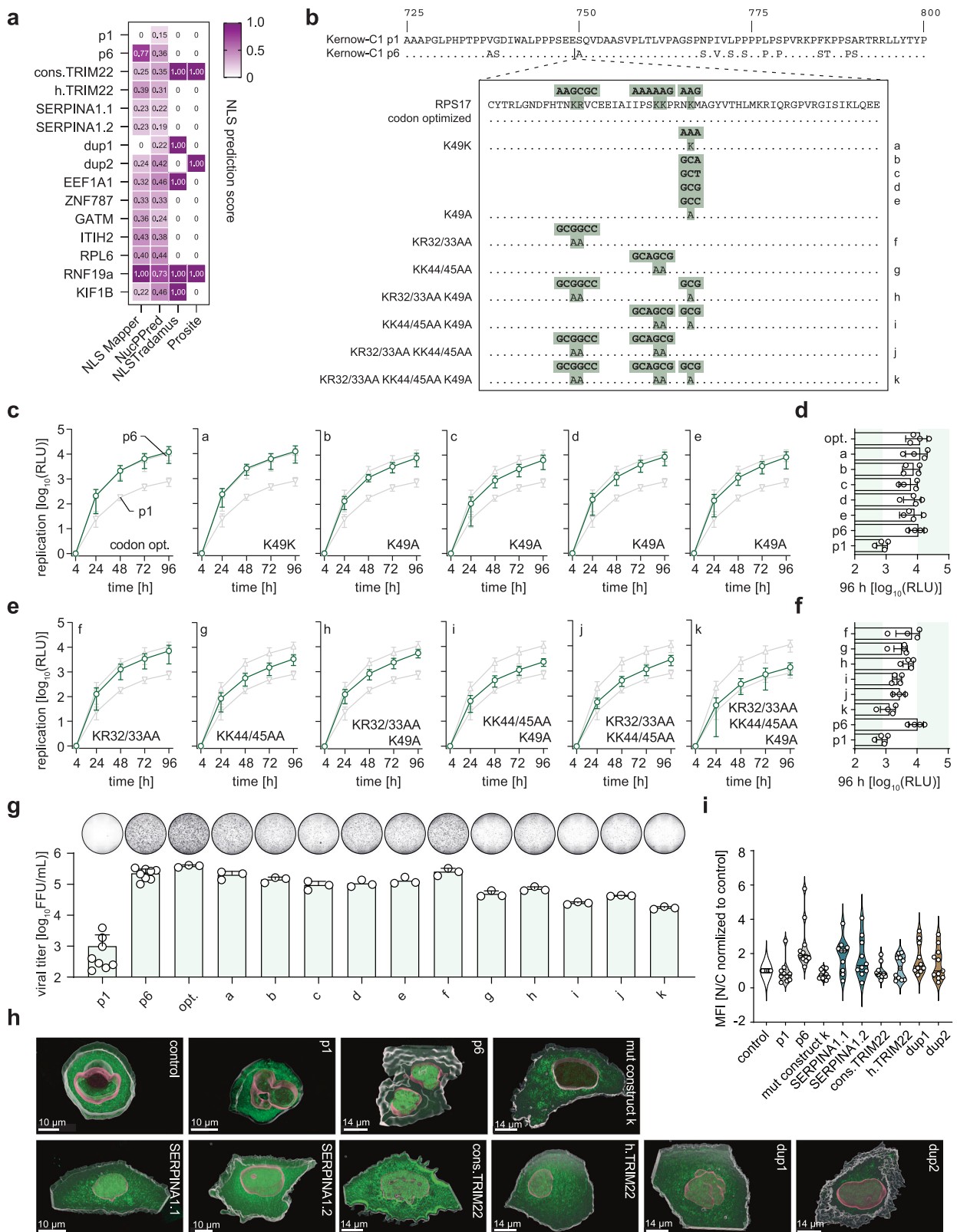

flexible linkers in p1 and p6, indicating that flexibility is no major determinant in the HVR for HEV replication. The NLS-based constructs could also not rescue the enhanced replication levels of p6. The different recombinant genomes replicated analogous to p1 (Fig. 6c), despite comparable predicted noodle-like secondary structures of the respective HVRs (Supplementary Movies 14–17). When engineering YFP constructs with trailing artificial insertions,

no disturbance of potential NLS (here only SV40) was observed via IF for the flexible or rigid linker (Fig. 6d, e and Supplementary Movies 18–25). To exclude the influence of sequence composition on replication, we constructed Gluc and YFP reporters with shuffled RPS17 insertions (Supplementary Fig. 4b, GenBank accession numbers PP408298-PP408301). In the p1 background, these shuffled constructs replicated only to extremely low levels, while both

**Fig. 5 | Nuclear localization signals as a common pattern of HVR insertions.**
**a** The insertion containing HVRs was analysed for the presence of NLS sequences or predicted nuclear localization by using the tools NLS Mapper (score normalized to 0 to 1). NucPred (Score 0 to 1), NLSTradamus (score 0 or 1) and Prosite (score 0 or 1). **b** Amino acid sequence of the RPS17 insertion from the Kernow-C1-p6 strain is shown. Alanine scanning mutagenesis was used to disrupt the NLS sequence. The generated constructs are shown in the alignment, Green areas above amino acids indicate codons. **c, e** The replicon system were used to analyse the impact of NLS mutants on the replication capacity of the Kernow-C1-p6 (p6) strain over time. Kernow-C1-p1 (p1) and p6 (grey triangles) were included as references while constructs of interest are depicted in green. Plotted are mean (+/- SD) relative light units (RLU) of $n = 3$ biologically independent experiments for construct j, $n = 4$ for other constructs, normalized to the four-hour value. **d, f** Respective 96-hour replication values are depicted as a column diagram. **g** The full-length system was

used to validate the impact of the NLS mutants on the virus production. A representative picture of a whole 96-well infected with non-enveloped HEVcc stained for the ORF2 protein (black) is shown above each column. Plotted are the means with standard deviation (+/- SD) of $n = 8$ biologically independent experiments for p1 and p6, $n = 3$ for other constructs. **h** Huh7 cells were transfected with plasmids encoding a triple eYFP alone (control) or in tandem with the HVRs of indicated constructs. The cells were fixed after 16-20 hours and mean fluorescence intensity (MFI) for eYFP was assessed for each compartment. See Supplementary. Movies 4–13. **i** The eYFP MFI was measured for each compartment for $n = 11$ cells per construct, individual outliers was removed applying ROUT ($Q = 1$) method implemented in GraphPad Prism. Depicted is the range of individual data points as violin plots with straight lines as median and dashed lines as quartiles. Source data are provided in Source Data file.

insertions, with and without a functional NLS (Supplementary Fig. 4c, d and Supplementary Movies 27–30) replicated to levels between p1 and p6 with 9.6-fold (SV40) and 16.1-fold (no NLS) reduction compared to p6 in the p6 context (Supplementary Fig. 4b). Interestingly, AlphaFold2 predictions foster the assumption, that differences in the 3D protein structure might mediate this phenotype (Supplementary Movie 26). In summary, an artificial insertion that mimics the sequence space of the HVR in either a rigid or flexible manner or different NLS sequences in that backbone could not recapitulate the enhanced replication fitness of HEV p6 or other patient-derived insertions.

### Viral determinants outside the HVR contribute to enhanced viral fitness

As the HVR protein space and NLS features did not fully recapitulate the replication capacity of p6, we next analyzed if determinants other than the insertion itself are responsible for the high viral fitness. In line with previously published data[20], we show that deletion of the *RPS17* insertion in p6 abrogates replication to p1 levels, while engineering the same insertion into p1 does not rescue HEV fitness (Supplementary Fig. 6). Comparing the amino acid sequences of ORF1 of p1 and p6 19 amino acid exchanges in addition to the inserted ribosomal *RPS17* RNA were identified (Fig. 7a). Of these substitutions only six amino acids were in greater distance to the insertion site (Fig. 7a), while thirteen substitutions near the insertion site (shown in Fig. 5b). Therefore, we mutated the p1 construct to contain the corresponding p6 amino acid for each position. The thirteen amino acids in proximity to the insertion site were cloned together with the *RPS17* insertion and were designated as *RPS17* + flanking regions (*RPS17*/FR). For five out of seven mutations, replication levels were comparable to p1 (Fig. 7b). However, the mutation A220T as well as the *RPS17*/FR mutant demonstrated a 8.6-fold and 3.6-fold increase in replication capacity, respectively (Fig. 7b). Subsequently, we cloned the combination of A220T and *RPS17*/FR, which added to a 81.7-fold increase in replication capacity, which could recapitulate the p6 replication levels (Fig. 7b, c). The full-length system confirmed the replication-enhancing effect of A220T and *RPS17*/FR alone as well as the combinatory effect (Fig. 7d) indicating that viral determinants outside the HVR were additionally required to rescue the p6 replication capacity. Next, we aimed to transfer the A220T and *RPS17*/FR phenotypes to another HEV-3 strain (83-2) that does not harbor any insertions[31]. The threonine at position 220 is already encoded in the 83-2 genome, therefore only the *RPS17* insertion and the *RPS17*/FR insertion were cloned into the 83-2 HVR (Fig. 7e, f). The *RPS17* insertion increased HEV replication only minimally, while the 83-2/*RPS17*/FR genome reached levels close to those of p6. However, this phenotype did not fully translate into the capacity to produce as many infectious particles as the p6 strain (Fig. 7g). In conclusion, these results show that the insertion-dependent enhancing replication fitness relies on the combination of genetic determinants outside the HVR with the HVR itself.

## Discussion

RNA viruses such as HEV diversify into populations with high intra-host variability providing a potential benefit to the virus population across changing environments (e.g. antiviral therapy). Recent reports identified several HEV strains harboring genomic rearrangements in tissue culture (e.g. Kernow-C1-p6) and in patients at the acute and chronic stages of infection indicating that enhanced population heterogeneity in the HVR is associated with viral fitness and adaptation[18,32–34]. In principle, sequences which are not required for virus replication are normally rapidly lost in small RNA viruses like HEV, implying a potential biological role of the HVR during HEV replication and/or pathogenesis[35]. However, the molecular mechanism behind this host-sequence snatching events with increased viral replication efficiencies remained unclear.

In serum samples of a chronically HEV-infected patient, two insertions of human origin, namely *TRIM22* and *SERPINA1* as well as duplications of the HVR sequence into the HVR were discovered. Furthermore, we analyzed seven additional insertions of human origin, which have been recently described[18]. First, we deleted the *RPS17* insertion from the strain p6, which in line with Emerson et al. results in replication levels similar to those of the parental strain p1, highlighting the role of this insertion as a determinant for viral fitness[20]. Next, we utilized a reverse-genetic HEV cell culture system[36] to characterize the impact of the identified insertions on viral replication, RBV sensitivity, and capacity to produce infectious particles by replacing the HVR of the strain p6 with the insertion containing HVRs. Of note, all thirteen analyzed insertions were able to increase replication and virus production. In the case of the here in vivo identified host-derived insertions, the abundance in the viral population did not correlate with replication efficacy, possibly arguing for directed selection. Other viruses like HIV-1 and HCV treatment resistance variants with mutations and insertions have been described[37–45]. Here, no change in treatment sensitivity to RBV could be observed, indicating that the insertions confer a fitness advantage and do not alter the sensitivity against RBV due to resistance mutations.

Utilizing multiple bioinformatic tools, we analyzed the HVRs of insertion-containing constructs for sequence or structural common patterns. First, we compared the amino acid frequencies of insertion containing HVRs to HVRs without insertions and found a 3-fold enrichment of the amino acid lysine and a predicated increase in the number of phosphoserines, O-linked glycosylation, acetylation as well as ubiquitination. These results are line in with previous studies of HEV-p6 and insertions identified in HEV patients[18,22,46]. The enrichment of those PTMs is an indicator of an evolutionary advantage of those variants and should be analyzed in future studies. Here, we focused on the conserved lysine residues, which were additionally linked to the selection of nuclear localization signals. Alanine scanning mutagenesis of the NLS in the *RPS17* insertion of the p6 strain revealed that lysine and arginine residues in the predicted NLS sequence are required for viral replication and efficient production of infectious particles. If

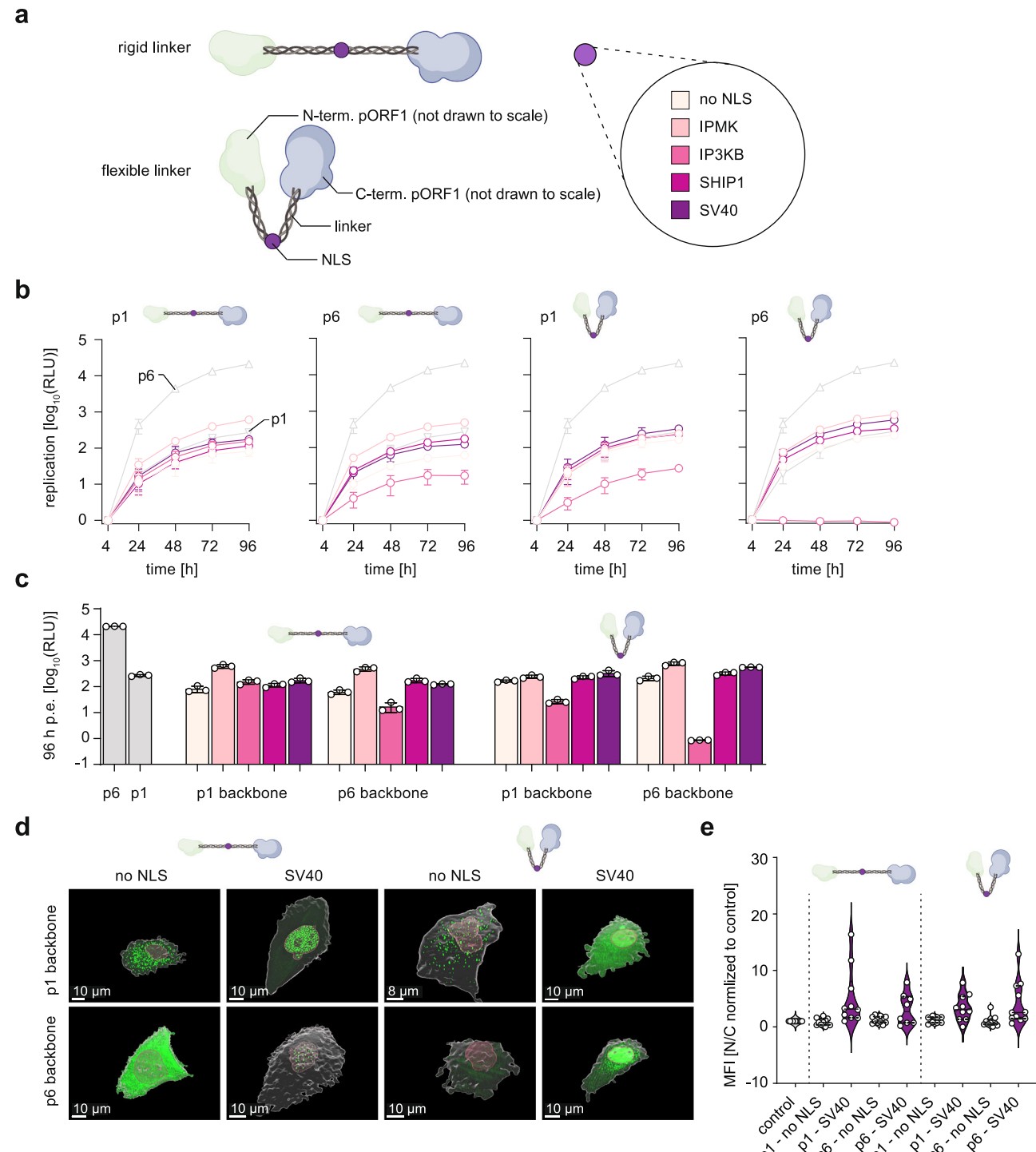

**Fig. 6 | Characterization of generated artificial insertions in the HEV HVR. a** Two types of artificial insertions were created and cloned in the Kernow-C1-p1 (p1) and Kernow-C1-p6 (p6) reporter replicon. A rigid XP linker, mimicking the high proline content of the HVR and one with a more flexible linker. Additionally, NLS sequences (IPMK, IP3KB, SHIP1, SV40) were incorporated in the middle of the artificial insertions. See Supplementary Movies 14–17. **b** HEV replication kinetics were measured with p1 and p6 as reference (both grey) while constructs of interest are depicted in purple shades. Plotted are the mean (+/- SD) relative light units (RLU) normalized to the four-hour value over time (hours post electroporation) of $n = 3$ biologically independent experiments. **c** For comparison the replication values 96 h p.t. of all constructs are plotted as column diagram.+/- SD **d** Huh7 cells were transfected with

plasmids encoding a triple eYFP in tandem with the artificial insertions including p1 and p6 flanking regions, respectively. Cells were fixed after 16–20 hours and the mean fluorescence intensity (MFI) for eYFP was measured for each compartment. Shown are example cells in 3D. eYFP is shown in green, the cell surface is depicted in white while the nuclear surface is depicted in red. See Supplementary Movies 18–25. **e** The eYFP MFI was measured for each compartment for ten $n = 11$ cells per construct, and individual outliers were removed by applying ROUT ($Q = 1$) method implemented in GraphPad Prism. Depicted is the range of individual data points as violin plots with a straight line as the median and dashed lines as quartiles. Source data are provided in the Source Data file.

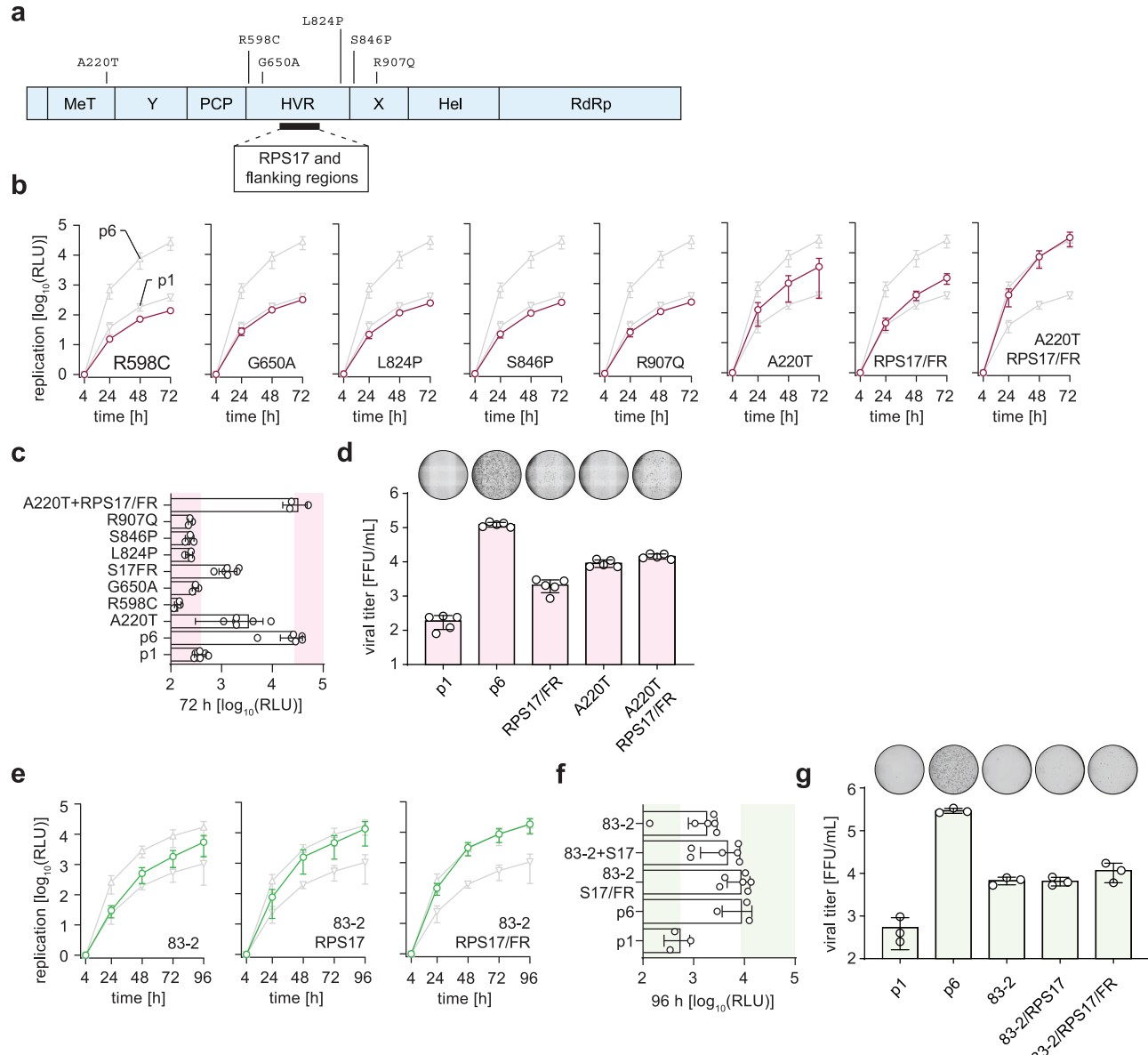

**Fig. 7 | Viral determinants for HEV replication in- and outside the HVR.**
**a** Depicted is the Kernow-C1-p1 (p1) ORF1 encoding sequences with mutations that differentiate it from Kernow-C1-p6 (p6) ORF1. Additionally, the RPS17 insertion site is indicated on the genome. The mutations over the genome were cloned separately into p1, while the thirteen variants near the RPS17 (see Fig. 5b) were cloned together with the *RPS17* RNA and were termed RPS17+flanking regions (RPS17/FR).
**b** Replication kinetics were measured with p1 and p6 as reference (grey triangles) while constructs of interest are depicted in red. Plotted are mean relative light units (RLU) normalized to the four-hour value over time (hours post electroporation) of $n = 3$ biologically independent experiments, $n = 6$ for A220T and RPS17/FR constructs. **c**, **f** The 72-hour replication values were plotted as a column diagram. The left red/green area depicts the p1 replication, while the right red/green area depicts the p6 replication level. **d** The HEV full-length system was used to produce infectious particles with indicated constructs, which were titrated onto HepG2/C3A cells to determine viral titers as FFU/mL via immunofluorescence. A representative

picture of a whole 96-well infected with non-enveloped HEVcc stained for the ORF2 protein (black) is shown above each column (mean, +/- SD). Dots represent individual data points of $n = 5$ individual experiments. **e** The RPS17 insertion alone or with a flanking region was cloned into the HEV3-83-2-27-Gluc replicon. Replication kinetics were measured with p1 and p6 as reference (grey triangles) while constructs of interest are depicted in green. Depicted are mean (+/- SD) relative light units (RLU) normalized to the four-hour value over time (hours post electroporation) of $n = 3$ biologically independent experiments for p1 and p6, $n = 6$ for other constructs. **g** The HEV full-length system was used to produce infectious particles with indicated constructs, which were titrated onto HepG2/C3A cells to determine viral titers as FFU/mL via immunofluorescence. A representative picture of a whole 96-well infected with non-enveloped HEVcc stained for the ORF2 protein (black) is shown above each column (mean, +/- SD). Dots represent individual data points of $n = 3$ individual experiments. Source data are provided in the Source Data file.

mutated completely, the replication level decreased to the value of the p1 strain and the amount of produced infectious particles was only 5.7% of the WT construct, which is in line with previous data[29]. However, this titer was still 32-fold enhanced compared to the construct p1 indicating that other viral determinants are also important. In line with a previous report, codon-based optimization further underlined the relevance of the protein sequence rather than possible RNA secondary

structure motifs[47]. Moreover, the translocation experiments could only show strong NLS activity for the p6 HVR, but not for *SERPINA1* and *TRIM22* insertions. The creation of artificial insertions attempting to mimic the HVR features showed that HEV tolerated artificial insertion with replication levels comparable to p1.

To test for genetic determinants outside the HVR, all 19 amino acid exchanges in ORF1 between p6 and p1 were cloned into p1 strain

to validate their impact on replication and virus production capacity. Of note, a single mutation in the ORF1 methyltransferase in combination with the *RPS17* insertion and its flanking regions reached replication levels as high as the p6 strain implying synergistically acting genetic elements of the insertion mechanism. This pro-viral effect of *RPS17*/FR was transferable to another HEV-3 strain naturally isolated without insertion.

In summary, these results highlight the existence and pro-viral effect of insertions in the HVR with unaltered sensitivity towards RBV. This pro-viral effect probably relies on several combinatory features like PTMs sites, NLS sequences and higher flexibility within the HVR as well as sequence-based determinants outside the HVR.

## Methods

### PCR amplification and clonal Sequencing

The patient was treated at Hannover Medical School, Germany. The study protocol was in line with the ethical guidelines of the Institutional Review Committee. The study was approved by the ethics committee of Hannover Medical School in Hannover, Germany (record 930–2011), and it conforms to the ethical guidelines of the 1975 Declaration of Helsinki. The patient gave written informed consent to participate in this study. RNA was isolated from 200 µl EDTA plasma of a chronically HEV-infected patient using the Cobas AmpliPrep total nucleic acid isolation kit (Roche, Switzerland) according to the manufacturer's instructions. Complementary DNA was synthesized using the Superscript™ IV Reverse transcriptase (Thermo Fisher Scientific) with random hexamers according to the manufacturer's instructions. The HVR region was amplified via nested PCR by using the Platinum™ Taq DNA polymerase (Thermo Fisher Scientific) and the two primers sets S-O-HEVpat-HVR (TGTGCTCGGGAATAAGACTTT) and A-O-HEVpat-HVR (AGGCTAACAGGGACTTGATATATA) (outer primer) as well as S-I-HEVpat-HVR (TTACCTATGAGCTCACCCCT) and A-I-HEVpat-HVR (CTTTGGGTTTTGCTCGACCT) (inner primer). The final PCR product was isolated via gel extraction (Gel and PCR Cleanup Kit, Macherey und Nagel) and utilized for TA cloning (pGEM®- Easy Vector System, Promega). The ligated vector was transformed into JM109 cells and blue-white screened for positive clones. The plasmids of 200 colonies were isolated and sent for Sanger sequencing. The Basic Local Alignment Search Tool (BLAST[48], was used to determine the HEV origin of the read and identify insertions. All insertions identified by clonal sequencing and reported by Lhomme et al.[18] were used to build multiple sequence alignments (MSAs) against the strains Kernow-C1-p1, Kernow-C1-p6 and 83-2 by using Clustal Omega[49]. Based on the MSAs the insertions were cloned into the Kernow-C1-p6 reverse genetic systems, thereby replacing the *RPS17* insertions and large parts of the HVR with the identified insertion containing HVRs.

### Phylogenetic analysis

Clonal sequencing data was used for phylogenetic analysis. A multiple sequence alignment was conducted using MAFFT[50], which was used to construct a maximum likelihood tree with 1000 bootstraps using iqTree2[51]. Tree visualization was performed in R using the libraries tidyverse, tidytree, ggtree, treeio and phylobase.

### Cell culture

HepG2 (ATCC Nr.: HB-8065, last cell line authentication 2023/07/05, Microsynth), Huh7 (last cell line authentication 2022/09/30, Microsynth) and Huh7.5 (kindly provided by Charles Rice, The Rockefeller University, New York, USA) were cultured in Dulbecco's Modified Eagle's Medium (DMEM, Invitrogen) supplemented with 10% fetal calf serum (FCS, GE Healthcare), 100 µg/mL streptomycin, 100 IU/mL penicillin (Invitrogen), 1%[vol/vol] nonessential amino acids (NEAA, Invitrogen) and 2 mM L-glutamine (Thermo Fisher Scientific). The HepG2 subclone C3A (HepG2/C3A, also kindly provided by Charles Rice, last cell line authentication 2023/07/05, Microsynth) was used for

infection experiments since greater infection efficiencies are achieved. HepG2/C3A cells were cultured in Eagle's minimum essential Medium (MEM, Gibco) supplemented with 10% [vol/vol] ultralow IgG FCS (Gibco, Cat. Nr.: 16250-078), 100 µg/mL gentamicin (Gibco), 2 mM L-glutamine (Thermo Fisher Scientific), 1 mM sodium pyruvate (Gibco), 1%[vol/vol] NEAAs (Gibco). HepG2 and HepG2/C3A cells were cultured on rat collagen-coated (SERVA) cell-culture dishes at 37 °C in 5% [vol/vol] $CO_2$ in an incubator.

### Plasmids

The plasmids encoding Kernow-C1-p6 (GT3; GenBank accession Nr.: JQ679013), Kernow-C1-p1(GT3; GenBank accession Nr.: JQ679014), and their derivatives with luciferase reporter were kindly provided by S.U. Emerson (NIAID). The plasmids encoding the strain HEV83-2-27 (GT3; GenBank accession Nr.: AB740232) and its *Gaussia* luciferase derivate (Gluc) were kindly provided by T. Wakita (National Institute of Infectious Diseases, Japan). All cloned mutations and insertions were performed by PCR-based mutagenesis, overlap-extension PCR and Gibson Assembly approaches. Further details regarding the cloning strategies and exact nucleotide sequences can be obtained upon request.

### In-vitro transcription and electroporation

All HEV encoding plasmids were used for in-vitro transcription by T7 polymerase followed by transfection into target cells via electroporation as previously described[24]. Briefly, plasmids were linearized by restriction digest with MluI and subsequently cleaned up using the QIAquick Spin Mini Kit (Qiagen). To produce viral genomic RNA, linearized plasmids were in vitro transcribed using a T7 polymerase with an additional capping step (m7G Cap analogue, Promega). Lastly, a DNase (Promega) was added to digest the initial plasmid construct. After RNA clean up (NucleoSpin RNA Clean-Up Kit, Macherey & Nagel), RNA integrity was determined by spectrophotometry and agarose gel electrophoresis.

*HEV replication and RBV sensitivity assay*

In-vitro transcribed RNA of the Gluc replicons was transfected into HepG2s cells by electroporation and 20,000 cells/well, were seeded in a collagen-coated 96-well plate in 100 µL DMEM complete medium in technical triplicates. The supernatants were collected at the indicated time points and stored at 4 °C until the measurement. To measure the sensitivity against RBV, the compound was solved in DMSO and added to the wells at the indicated final concentrations ranging from 100 µM to 0.097 µM. The supernatant of RBV-treated cells was harvested after 96 h. Gluc luciferase activity was measured as previously described[52]. In brief, 20 µL of the cell culture supernatant was added per well on a LUMITRAC 600 96-Well plate, followed by automated addition of coelenterazine substrate and detection of luminescence using a Centro XS[3] LB 960 luminometer (Berthold Technologies). Dose-response curves were calculated using a four-parameter log-logistic method implemented in Graph Pad Prism v10.

### Production of cell-culture-derived HEV (HEVcc)

Full-length HEV transcripts were produced as described above and transfected into HepG2 cells via electroporation. After electroporation, $5 \times 10^6$ cells were seeded onto a collagen-coated 10-cm plate and cultured for seven days at 37 °C. Then the cell culture supernatant, containing the enveloped HEVcc was harvested and filtered through a 0.45 µm filter to remove cell debris. The electroporated cells were trypsinized and the single-cell suspension was centrifuged $200 \times g$ for 5 min. The cell pellet was resuspended 1.6 mL DMEM and the cells were lysed by three freeze-thaw cycles to release the intracellular non-enveloped HEVcc. The virus suspension was centrifuged at $10,000 \times g$ for 10 min to remove cell debris. The virus particles were stored at −80 °C and thawed and kept on ice until being used.

## Virus titration

A serial dilution of the produced HEVcc on HepG2/C3A was performed to determine the infectious titer. To do so, $2 \times 10^4$ cells/well were seeded on a 96-well plate in technical duplicates one day prior to titrating. On the next day, HEVcc was serially diluted fivefold (intracellular, non-enveloped HEVcc). The cells were incubated for seven days post-infection, washed with PBS and fixed with 3% (w/vol) paraformaldehyde (PFA) for 20 min. The fixed cells were then subjected to immunofluorescence staining against the capsid protein ORF2 as detailed in the following section and the number of focus forming units (FFU) was counted and calculated as described in Meister et al.[36].

## Immunofluorescence staining

Cells were fixed with paraformaldehyde and subsequently permeabilized with 0.2% (vol/vol) Triton X-100 in PBS for 5 min at room temperature followed by three PBS washes. Then the cells were blocked with 5% (vol/vol) horse serum in PBS for 1 h at room temperature (RT). The ORF2 protein was stained with a polyclonal HEV-3 capsid-specific rabbit serum (1:5000 in 5% horse serum, kind gift of R.G. Ulrich, Friedrich Loeffler Institute, Germany) overnight at 4 °C. Cells were washed thrice with PBS to remove unbound antibody and goat anti-rabbit antibody (Alexa Fluor 488, 1:1000 in 5% horse serum, Invitrogen, catalog number: A11008, lot: 2743033) was added for 1 h at RT in the dark. Subsequently, the secondary antibody was removed by three PBS washes and the nuclei were stained with 4',6'-diamidino-2-phenylidole (DAPI, 1:10,000 in $H_2O$) for 5 min. Finally, the cells were washed with PBS and stored at 4 °C in the dark until imaging. A Keyence BZX800 microscope was utilized for image acquisition in 4x magnification. Whole wells were imaged and stitched into a single image using FIJI[53].

## Detection of *TRIM22* mRNA

Cellular total RNAs of HEV p6 full-length or yeast tRNA (Sigma-Aldrich) transfected HepG2 or Huh7.5 cells were extracted using a NucleoSpin RNA kit (Macherey-Nagel, Düren, Germany) with the involvement of RNase-free DNase, following the manufacturer's protocol. Total RNAs were reverse transcribed using a PrimeScript 1st strand cDNA synthesis kit (TaKaRa Bio, Otsu, Japan). Quantitative PCR was carried out using 400 nmol of primers together with SYBR Premix Ex Taq (TaKaRa Bio) and the LightCycler 480 system (Roche, Basel, Switzerland). Primer sequences for amplification of HEV RNA[54] and h*TRIM22*[55] have been described previously, and hGAPDH served as a reference gene to normalize gene expression[56]. The relative gene expression was determined using the cycle threshold (ΔΔCT) method.

## Fluorophore translocation assay

The fluorophore translocation assay described by Kenny et al.[29]) was adapted to screen the insertion containing HVRs for the presence of functionally active nuclear localization sequences. The HVRs were PCR amplified using the primer spk437 (CATCATCTCGAGATGCCAGAGCAGTATGTCCTGTC), spk438 (CATCATAAGCTTCAGCCAATCACAGTCTGATTCAAA) and spk438-modified (CATCATAAGCTTCAGCCAGTCGCAGTCAGACTCAAA) and cloned into the eYFP3 vector by restriction ligation cloning utilizing the enzymes XhoI and HindIII. The primer combination spk437 and spk438 was used for the *TRIM22* containing HVRs while a modified primer SPK438-modified had to be used for *SERPINA1* and duplication containing HVRs due to four mutations. The generated plasmids were transfected into Huh7 cells using Lipofectamine 3000 (Thermo Fisher Scientific) according to the manufacturer's instructions. In brief, $10 \times 10^5$ cells were seeded on collagen-coated coverslips in 24-well plates using 400 µL of growth medium. On the following day, 25 µL Opti-MEM was mixed with 0.75 µL Lipofectamine 3000. In a second reaction tube, 25 µL Opti-MEM was mixed with 250 ng of the eYFP3 plasmid and 0.5 µL P3000 reagent. The second tube was preincubated for 2 min and mixed with the first tube. The mixture

was incubated for 15 min and 50 µL were added per well onto the Huh7 cells. The cells were fixed after 16-20 h and subjected to immunofluorescence staining of the tubulin cytoskeleton (alpha Tubulin B-7, 1:1,000 in 5% horse serum, Santa Cruz Biotechnology, catalog number: sc-5286, lot: H2719) and the nucleus (NucSpot® Live 650, Biotium). The coverslips were mounted on glass slides and analyzed using a Zeiss Elyra 7 microscope with a 63x oil immersion objective (Plan-Apochromat 63x/1.4 Oil DIC, Carl Zeiss Microscopy GmbH, Germany). A lattice-structured illumination (Lattice-SIM) with 13 phases of the SIM grid was used to create super-resolution images. ZEN black 3.0 R was employed to process the raw confocal SIM images. The nuclear and cytoplasmic surfaces were reconstructed volumetrically and the mean eYFP fluorescence intensity (MFI) was measured in each compartment using the Imaris 10.0.1 (Oxford Instruments, UK) surface function. The ratio of nuclear to cytoplasmic MFI was used as a surrogate for NLS strength.

## In-silico analysis of insertion sequences

The amino acid sequence of the HVRs in the boundaries as defined by Muñoz-Chimeno et al.[14] was used for the *in-silico* analysis of PTMs, nuclear localization signals, and the differences in the amino acid composition of insertion containing HVRs. The amino acid composition of insertion containing HVRs was analyzed using a composition profiler with 10,000 bootstrap iterations and Bonferroni correction[57]. Therefore, HEV-3 full-length sequences were downloaded from HEV-Glue (http://hev.glue.cvr.ac.uk/). Multiple sequence alignments versus reference strains Kernow-C1-p1 and HEV3-83-2-27 were constructed using Clustal Omega[49]. All insertion-containing sequences were removed manually and the remaining 289 HEV HVR sequences were used as reference. Posttranslational modifications (PTM) were predicted using musite[58] GPS-Pail[59] and BDM-PUB[60]. The NLS sequence were predicted by NLSmapper[61], NucPred[62], NLSTradamus[63] and Prosite[64]. Single cell sequencing data of the liver cell atlas[23] was analyzed for the expression level of the inserted genes in various liver cell types. The protein structure of insertion-containing constructs was predicted by using ColabFold v1.5.2[65]. Therefore, the amino acid sequence of all constructs ranging from the 5' of the PCP domain to the 3' of the helicase was used. The generated PDB files were analyzed in PyMol 2.5[66].

## Reporting summary

Further information on research design is available in the Nature Portfolio Reporting Summary linked to this article.

## Data availability

Data that support the findings of this study have been deposited in GenBank with the accession codes OR726668-OR726848, OR700721-OR700740, and PP408296-PP408301 [example: https://www.ncbi.nlm.nih.gov/nuccore/OR726668]. Expression data of HEV infected PHH are available at GEO database under accession no. GSE135619. Expression data of the liver atlas are available at GEO database under accession no. GSE192742. All other relevant Source data are provided with this paper.

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

## Acknowledgements

Confocal laser-scanning and SR-SIM microscopy instruments were funded by the DFG and the State Government of North Rhine-Westphalia (INST 213/840-1 FUGG). D.T. was supported by grants from the Deutsche Forschungsgemeinschaft (DFG, German Research Foundation, project no. 448974291) and by the German Federal Ministry of Education and Research (BMBF, project: VirBio, Grant number: 01KI2106). E.S. was supported by grants from the German Federal Ministry of Health (ZMVI1-2518FSB705), the German Centre for Infection Research (DZIF, TTU 05.823_00), the German Federal Ministry of Education and Research (BMBF, project HepEDiaSeq — FKZ 01EK2106A), the German Research Foundation (DFG, project no. 510558817) and by the National Institutes of Health (NIH R21AI151736). K.F.W. was supported by the German Research Foundation (DFG) under Germany's Excellence Strategy - EXC 2033 - 390677874 - RESOLV. We thank all members of the Department for Molecular and Medical Virology at the Ruhr University Bochum for helpful suggestions and discussions, R.G. Ulrich for providing the ORF2 specific rabbit serum, S.U. Emerson for providing the constructs Kernow-C1-p6 and -p1 and T. Wakita for providing the construct HEV3-83-2-27. We further thank C. M. Rice for providing Huh7 and Huh7.5 cells.

## Author contributions

C.T.B., A.P., S.P.K., K.F.W., P.B., H.W., E.S., and D.T. provided substantial contributions to study conception and analysis design. M.H.W., T.L.M., M.M., L. K., A.S., and D.T. contributed to data acquisition. M.H.W., M.K.N., A.G., Y.B., V.B., E.S., and D.T. participated in data analysis and interpretation. The article was drafted by E.S. and D.T. and revised critically for important intellectual content by M.H.W., C.T.B., A.P., S.P.K., K.F.W., P.B., and H.W.

## Funding

## Competing interests

The authors declare no competing interests.

## Additional information

[1]Department for Molecular and Medical Medicine, Ruhr University Bochum, Bochum, Germany. [2]Institute for Infection Research and Vaccine Development (IIRVD), Centre for Internal Medicine, University Medical Centre Hamburg-Eppendorf (UKE), Hamburg, Germany. [3]Department for Clinical Immunology of Infectious Diseases, Bernhard Nocht Institute for Tropical Medicine (BNITM), Hamburg, Germany. [4]German Centre for Infection Research (DZIF), Partner site Hamburg-Lübeck-Borstel-Riems, Hamburg, Germany. [5]European Virus Bioinformatics Center (EVBC), Jena, Germany. [6]Institute for Experimental Virology, TWINCORE Centre for Experimental and Clinical Infection Research, a Joint Venture between the Medical School Hannover (MHH) and the Helmholtz Centre for Infection Research (HZI), Hannover, Germany. [7]Hannover Medical School, Institute for Medical Microbiology and Hospital Epidemiology, Hannover, Germany. [8]Department of Molecular Cell Biology, Institute of Biochemistry and Pathobiochemistry, Ruhr University Bochum, Bochum, Germany. [9]Department of Biochemistry of Neuro-degenerative Diseases, Institute of Biochemistry and Pathobiochemistry, Ruhr University Bochum, Bochum, Germany. [10]Division of Viral Gastroenteritis and Hepatitis Pathogens and Enteroviruses, Department of Infectious Diseases, Robert Koch Institute, Berlin, Germany. [11]Department of Molecular Biology, Princeton University, Princeton, NJ, USA. [12]Center for Food Animal Health, Departments of Animal Sciences and Veterinary Preventive Medicine, The Ohio State University, Wooster, OH 43210, USA. [13]Cluster of Excellence RESOLV, Bochum, Germany. [14]Department of Gastroenterology, Hepatology and Endocrinology, Hannover Medical School, Hannover, Germany. [15]German Center for Infectious Disease Research (DZIF); Partner Sites Hannover-Braunschweig, Braunschweig, Germany. [16]Excellence Cluster 2155 RESIST, Hannover Medical School, Hannover, Germany, Braunschweig, Germany. [17]German Centre for Infection Research (DZIF), External Partner Site, Bochum, Germany. ✉e-mail: Eike.Steinmann@ruhr-uni-bochum.de; Daniel.Todt@ruhr-uni-bochum.de

