## [Peer Review File · Nature Communications]

Editorial Note: Parts of this peer review file have been redacted as indicated to maintain the confidentiality unpublished data.REVIEWER COMMENTS

Reviewer #1 (Remarks to the Author):

Wissing et al. are submitting a manuscript which provides a detailed molecular characterization of sequence insertions found in the hepatitis E virus (HEV) genome of infected patients. Authors' study started from a clinical case where they identified different types of sequence insertions within the hypervariable region (HVR) of HEV genome. Sequence insertion in the HVR was already described in the past, including by the authors, and was supposedly favorable for viral replication similarly to what observed with the prototype p6 molecular clone. This study is the first one investigating the potential replication advantage acquired by sequence insertion in HVR. To this, Wissing and colleagues used the p6 clone in which they switched the RPS17 insertion in its HVR by the newly identified insertions as well as others described in the literature. Data generated by the authors showed an enhanced replication with all constructs engineered. Very systematically they also looked at infectivity of viral progeny in recombinant full-length genomes. To address what are the advantages acquired by the virus with these insertions, the authors performed *in silico* analyses to look for specific amino acid composition as well as to identify potential posttranslational modifications. As identified earlier, there is an enrichment in positively charged residues and nuclear localization signals (NLS) are frequently predicted. Mutagenesis approach combined with replication assay demonstrated the importance of some lysine residues present in the RPS17 insertion found in p6. Moreover, the authors showed, using a fluorescence microscopy assay, that these lysine are involved in a functional NLS and further demonstrated that other sequence insertions harbor a NLS. In their efforts to confirm the potential role of these insertions as NLS as well as linker region, Wissing and colleagues prepared replicon constructs harboring either rigid or flexible linker with a variety of known NLS. The replication results showed that neither the type of linker nor the NLS could reproduce the RPS17 advantage. Lastly, the authors provided a detailed functional characterization of the adaptive changes selected in p6 as compared to p1 clone. They could demonstrate that beyond the RPS17 insertion in the HVR, the aa substitution A220T contribute significantly to the enhanced replication observed with p6. Replication enhancing effect of the RPS17 could be further demonstrated by introducing it in another GT3 clone which does not harbor any insertion.

Overall, this study represents an important source of information regarding insertion found in patients. Often observed, these insertions were never functionally tested in such details. Therefore, this work represents important piece of data on which future observations of insertions in the HVR of HEV genome may scaffold hypotheses.

Major issues:

1- If correctly understood, all constructs harboring the different insertions identified in patients were done in the p6 backbone where these insertions replaced the RPS17 sequence and the flanking regions. In the replication assay, kinetic data obtained with these constructs were shown as

compared to the ones of p6 and p1, as controls. However, as we learned later in the manuscript, in Fig 7, other determinants are contributing to increase the replication capacity, such as A220T. Given that this latter mutation is already significantly contributing to the increase replication observed with p6 compared to p1, the results presented in Figures 2, 3 and 5 may appear misleading. A comparison to p1+A220T would be more appropriate, as the other mutations did not show any impact on replication. For example, it would be beneficial to have this control in Figure 5G where it may more clearly demonstrate the importance of the lysine residues present in the insertion.

Complementary to this point, the manuscript may benefit from another structure. Indeed, as the study focused on insertions in HVR, I would suggest to move Fig 7 (results on the viral determinants) earlier in the manuscript (as a Fig 2 for example). It may offer the advantage to establish the rationale for swapping the insertions in HVR based on p6 clone. This re-organization should allow to then focus on the characterization of the HVR insertions.

2- At this stage, results showed in Figure 6 are not very conclusive. The low replication level of the replicons harboring linkers and various NLS tend to show that these insertions are selected beyond their "spacer" and NLS functions. An important control experiment which may be added is the analysis of these constructs in the context of fluorescence microscopy assay to observe YFP targeting to the nucleus (system described in Fig 5H). Furthermore, the sequences used as linkers are rather artificial and have a sequence composition somewhat different than found usually in insertions. To overcome these limitations, did the authors consider to introduce a shuffled sequence of RPS17, with and without the known NLS, to exclude that length and amino acid composition are important determinants?

Minor issues:

1- Other insertions have been described in the literature, especially some with HEV sequences, e.g. RdRp. The authors may want to extend their *in silico* analyses with these sequences as well.

2- L193 p9, "Gaussian" should read "Gaussia"

3- L260 p11, "TRIM-22" should read "TRIM22"

4- L277 p12, "The number of those PTMs was increased by up to 13 modifications (O-linked glycosylation for duplication 1) in this case on an HVR length of only 156 AA", the authors may want to modify the sentence as follow: "The number of those predicted PTMs was increased by up to 13 (O-linked glycosylation for duplication 1) in this case on an HVR length of only 156 AA".

5- L362 p15, the authors may want to add few words on the composition of the flexible linker in the text.

6- Fig 6C, labels for p6 and p1 below the corresponding bars are missing.

7- Fig 6B, the scheme showing the PCP and X domains connected by either rigid or flexible linker may introduced some confusion as the linker is inserted within the entire ORF1 protein, in p1 or p6 replicon construct.

8- As a conclusion of the results shown in Fig 6, the authors stated that rigid linkers led to lower replication level than flexible ones (I368-370). It does not look so clear from the data presented. The authors may want to clarify this point.

9- Fig 7A, label of the polymerase region in the scheme indicates "RpRp" instead of "RdRp".

Reviewer #2 (Remarks to the Author):

This study analyzed host and virus-derived insertions in the hypervariable region (HVR) of the HEV ORF1 polyprotein. While these genomic rearrangements associated with an increased HEV replication have been characterized by different teams the underlying molecular and cellular mechanisms involved in this phenomenon are unknown.

The authors used different constructions based on the introduction of insertions in a luciferase replicon and a full-length viral genome of the Kernow-C1-p6 strain. They also tested the effect of these insertions on ribavirin antiviral activity.

Despite well-conducted experiments, the authors failed to identify a common mechanism involved in the enhanced replication conferred by these insertions and suggest that combinatory factors previously suspected like post translational modifications sites, NLS sequences and HVR flexibility explain the pro-viral effect.

The authors also showed that increased replication due to insertions did not affect the susceptibility to ribavirin.

Comments :

1. Nucleotide insertions of viral and host origin have been described in the ORF1 HVR of the HEV genome in several studies including Shukla PNAS 2011 ; Shukla J Virol 2012 ; Nguyen J Gen Virol 2012 ; Lhomme J Virol 2014 ; Lhomme Frontiers in Microbiol 2020 ; Papp Scientific reports 2022 ; Biederman Liver international 2022. In this study, 1 viral-derived insertion (major variant, 52.4%) and 2 host-derived insertions (minority variants, 3.2% and 10.7%) were identified from a single chronically infected patient previously treated with ribavirin. Did the authors screened other patients

with a similar clinical profile to estimate the frequency of these genetic rearrangements or insertions ?

2. After genetic engineering of viral and host-derived insertions (TRIM22, SERPINA1) into the Kernow-c1 p6 backbone, is there a relationship between the replicative capacity in vitro of these constructions and the proportions of variants identified in vivo ? This point should be discussed.

3. Only a fragment of host gene is inserted in HVR (for instance a fragment of TRIM22 or SERPINA1). Is there a similar effect on the replicative capacity if another fragment (same length) of the same gene is inserted ?

4. Data obtained with the replicon system show that increased replication efficiency did not affect ribavirin susceptibility. It is interesting to note that a similar observation was made with polymerase mutations associated to ribavirin therapy. The mechanism of action of ribavirin on HEV is uncertain and no key mutations associated to ribavirin resistance has been characterized.

5. Characterization of databank-derived HEV insertions

Line 220 : several insertions characterized by Lhomme in the HVR including the host insertion ITIH were reported for the first time in J Virol 2014, the others in Frontiers microbiology 2020.

Line 224 : The insertion GATM was detected after replication in a cell culture systems, not in vivo.

6. A crucial point is to understand the mechanism for the observed replication advantage conferred by viral and host-derived insertions in the HVR. The use of AlphaFold2 to analyze the impact of insertions on HVR could be inappropriate because HVR is an intrinsically disordered region (Purdy PLoS One 2012). Therefore the low prediction confidence score observed for the HVR with or without insertions was expected.

7. The influence of most viral and host-derived insertions described in the present manuscript on post-translational modifications has been previously studied using different bioinformatic tools. In addition, Kenney and Meng showed that Lysine residues in insertions were important for enhanced virus replication (Kenney J Virol 2015).

8. Lines 307-334 : the added value of the alanine scanning of the NLS sequence in the RPS17 insertion of p6 is not clear because these experiments were previously performed (Kenney J Virol

2015). The importance of the protein sequence of the insertions rather than the RNA sequence/structure has been previously reported.

9. The 19 aminoacid positions that are different between p1 and p6 (in addition to the inserted RPS17) could play a role in HEV replication. The authors showed that the mutation A220T in the methyltransferase region and HVR mutations in close proximity to the insertion site could restore a replication similar to p6 with the replicon system (but not virus production with the recombinant full-length genome). Are these observations described for Kernow strain generalizable ?

Reviewer #3 (Remarks to the Author):

MINOR REVISION

The manuscript written by Wißing et al. on Genetic determinants of host-and virus- derived insertions for hepatitis E virus replication is very interesting and innovative, providing insights in understanding the adaptation potential of circulating HEV strains and genetic determinants for viral replication. It deserves publication. All bioinformatic analyses and in vitro experiments are well described and performed.

However, some minor adjustments should be addressed before.

1) The most abundant type of rearrangement were duplications of HEV sequences (98 clones, 52.4 %), mostly of the HVR into the HVR.

In figure 1 and in the text: for duplication 1 and duplication 2, it would be good to add in the panel D the information of the location of the original sequence that is duplicated (N aa from position X to position Y)

2) In figure 5, regarding the synonymous mutant a (K49K AAG ->AAA) why in panel H the plasmid with mutant a Lysin K does not enter in the nucleus? It should be the K49A mutant? This experiment has been done? Please describe better this part.

RUHR-UNIVERSITÄT BOCHUM | 44801 Bochum | Germany
Dr. Daniel Todt | Universitätsstr. 150 | MA6/40

MEDIZINISCHE FAKULTÄT

To the
Reviewer

**Institut für Hygiene und Mikrobiologie
Abteilung für Molekulare &
Medizinische Virologie**
Gebäude MA 6/40
Universitätsstraße 150, 44801 Bochum
Prof. Eike Steinmann
Dr. Daniel Todt
Fon +49 (0)234 32-22463
Fax +49 (0)234 32-14797
eike.steinmann@rub.de
daniel.todt@rub.de
www.rub.de/virologie

March 2024

Response letter accompanying Revision NCOMMS-23-47687A

Dear Reviewer,

thank you for your critical reading and valued input of our submitted manuscript entitled “**Genetic determinants of host-and virus-derived insertions for hepatitis E virus replication**”. We were pleased to learn that you found our work interesting.

We now provide a revised version of the manuscript that carefully considers the issues raised by you. Please find the detailed response and explanation to your questions below.

Thank you for your time and effort. We appreciate the time you invested to help us improve our manuscript.

Sincerely yours,

Dr. Daniel Todt and Prof. Eike Steinmann

In the following we address each of the reviewer’s comments (given in *italics*). We provide two versions of our revision, one in which all changes are marked and one ‘clean’ version.

Reviewer 1:

Reviewer #1 (Remarks to the Author):

Wissing et al. are submitting a manuscript which provides a detailed molecular characterization of sequence insertions found in the hepatitis E virus (HEV) genome of infected patients. Authors' study started from a clinical case where they identified different types of sequence insertions within the hypervariable region (HVR) of HEV genome. Sequence insertion in the HVR was already described in the past, including by the authors, and was supposedly favorable for viral replication similarly to what observed with the prototype p6 molecular clone. This study is the first one investigating the potential replication advantage acquired by sequence insertion in HVR. To this, Wissing and colleagues used the p6 clone in which they switched the RPS17 insertion in its HVR by the newly identified insertions as well as others described in the literature. Data generated by the authors showed an enhanced replication with all constructs engineered. Very systematically they also looked at infectivity of viral progeny in recombinant full-length genomes. To address what are the advantages acquired by the virus with these insertions, the authors performed in silico analyses to look for specific amino acid composition as well as to identify potential posttranslational modifications. As identified earlier, there is an enrichment in positively charged residues and nuclear localization signals (NLS) are frequently predicted. Mutagenesis approach combined with replication assay demonstrated the importance of some lysine residues present in the RPS17 insertion found in p6. Moreover, the authors showed, using a fluorescence microscopy assay, that these lysine are involved in a functional NLS and further demonstrated that other sequence insertions harbor a NLS. In their efforts to confirm the potential role of these insertions as NLS as well as linker region, Wissing and colleagues prepared replicon constructs harboring either rigid or flexible linker with a variety of known NLS. The replication results showed that neither the type of linker nor the NLS could reproduce the RPS17 advantage. Lastly, the authors provided a detailed functional characterization of the adaptive changes selected in p6 as compared to p1 clone. They could demonstrate that beyond the RPS17 insertion in the HVR, the aa substitution A220T contribute significantly to the enhanced replication observed with p6. Replication enhancing effect of the RPS17 could be further demonstrated by introducing it in another GT3 clone which does not harbor any insertion.

Overall, this study represents an important source of information regarding insertion found in patients. Often observed, these insertions were never functionally tested in such details. Therefore, this work represents important piece of data on which future observations of insertions in the HVR of HEV genome may scaffold hypotheses.

We would like to thank the reviewer for the positive comment and appreciate the time invested to help improve our manuscript.

Major issues:

1- If correctly understood, all constructs harboring the different insertions identified in patients were done in the p6 backbone where these insertions replaced the RPS17 sequence and the flanking regions. In the replication assay, kinetic data obtained with these constructs were shown as compared to the ones of p6 and p1, as controls. However, as we learned later in the manuscript, in Fig 7, other determinants are contributing to increase the replication capacity, such as A220T. Given that this latter mutation is already significantly contributing to the increase replication observed with p6 compared to p1, the results presented in Figures 2, 3 and 5 may appear misleading. A comparison to p1+A220T would be more appropriated, as the others mutations did not show any impact on replication. For example, it would be

beneficial to have this control in Figure 5G where it may more clearly demonstrate the importance of the lysine residues present in the insertion.

Complementary to this point, the manuscript may benefit from another structure. Indeed, as the study focused on insertions in HVR, I would suggest to move Fig 7 (results on the viral determinants) earlier in the manuscript (as a Fig 2 for example). It may offer the advantage to establish the rationale for swapping the insertions in HVR based on p6 clone. This re-organization should allow to then focus on the characterization of the HVR insertions.

We thank this referee for the diligent evaluation of our study and for raising this point. It is correct that the insertions were all generated and tested in the p6 backbone, where they replaced the RPS17 sequence including the flanking regions. As both, the insertion with the flanking regions as well as the A220T mutation contribute to the increase replication fitness, we would like to keep the flow of the manuscript with the identification of A220T at the end (if the reviewer agrees). In our view, the combinatory effect of both determinants for HEV replication is better displayed, when showing the A220T mutation at the end. To not mislead the reader as the reviewer anticipated, we now changed the wording in the results section for Fig. 2, 3 and 5 to the comparison of the novel insertions with p6 and between them. We hope that the reviewer agrees with this feedback.

2- At this stage, results showed in Figure 6 are not very conclusive. The low replication level of the replicons harboring linkers and various NLS tend to show that these insertions are selected beyond their “spacer“ and NLS functions. An important control experiment which may be added is the analysis of these constructs in the context of fluorescence microscopy assay to observe YFP targeting to the nucleus (system described in Fig 5H). Furthermore, the sequences used as linkers are rather artificial and have a sequence composition somewhat different than found usually in insertions. To overcome these limitations, did the authors consider to introduce a shuffled sequence of RPS17, with and without the known NLS, to exclude that length and amino acid composition are important determinants?

As suggested, we now analyzed the different NLS constructs in the context of the YFP-based IF microscopy assay (suppl. Videos S24-31). We present the results in the novel Figures 6D and E. As concluded from the images as well as from the ratio of mean fluorescent intensities, all SV40 NLS harboring constructs predominantly localize to the nucleus, while for the other construct without NLS no preferential nuclear localization, or signals only in the cytoplasm is observed. We agree that these constructs do not replicate to levels of the parental p6 strain (Figure 6B), nevertheless, except for one construct, RNA replication is also not completely abrogated. Interestingly, when shuffling the RPS17 insertion, as suggested by the reviewer, we observe reduced replication compared to the respective parental strains in the p6 and the p1 background independent of a functional NLS (suppl. Fig. 4B-D). We concluded, as already anticipated by the reviewer, that determinants beyond amino acid composition and length of the insertion are essential. Our newly presented alphafold predictions (suppl. Video S32) point to rearrangements in the 3D structure of the HVR harboring shuffled RPS17 insertions that might be responsible for differences in replication capacity. Nuclear localization of YFP constructs harboring functional NLS was given also in the context of the shuffled RPS17 insert (suppl. Videos S33-36).

Minor issues:

1- Other insertions have been described in the literature, especially some with HEV sequences, e.g. RdRp. The authors may want to extend their in-silico analyses with these sequences as well.

Thank you for that suggestion. We expanded our in-silico analysis to the strain HEV 47832c, which has an insertion of the RdRp in its HVR. The analysis shows that the RdRp insertion in HVR has a similar profile of amino acid usage, NLS prediction and PTM prediction like the p6 strain and the other identified insertions as shown in response letter Figure 1. Since we did not include this insertion in our in vitro assays, we would prefer to not include it in the manuscript and present it only in the response letter.

Response letter Figure 1: In silico analysis of the HEV 47832c strain with RdRp insertion. A) Presented are the amino acid composition of insertion containing HVRs vs. non-insertion containing HVRs. The predicted number of various B) PTMs and C) NLS strength was analyzed for the HEV 47832c strain using the tools described in the manuscript.

2- L193 p9, "Gaussian" should read "Gaussia"

Now corrected.

3- L260 p11, "TRIM-22" should read "TRIM22"

Now corrected.

4- L277 p12, "The number of those PTMs was increased by up to 13 modifications (O-linked glycosylation for duplication 1) in this case on an HVR length of only 156 AA", the authors may want to modify the sentence as follow: "The number of those predicted PTMs was increased by up to 13 (O-linked glycosylation for duplication 1) in this case on an HVR length of only 156 AA".

We modified the sentence as suggested.

5- L362 p15, the authors may want to add few words on the composition of the flexible linker in the text.

As suggested, we now added more information about the flexible linker, namely its serine- and glycine-rich composition (page 16, line 386 in the tracked version).

6- Fig 6C, labels for p6 and p1 below the corresponding bars are missing.

Now corrected.

7- Fig 6B, the scheme showing the PCP and X domains connected by either rigid or flexible linker may introduced some confusion as the linker is inserted within the entire ORF1 protein, in p1 or p6 replicon construct.

Thank you. We now modified the scheme in Fig. 6A with the terms “N-term. pORF1” and “C-term. pORF1” accordingly.

8- As a conclusion of the results shown in Fig 6, the authors stated that rigid linkers led to lower replication level than flexible ones (l368-370). It does not look so clear from the data presented. The authors may want to clarify this point.

Thank you for that comment. We now modified the sentence as the rigid and flexible linker constructs replicated to similar degrees (page 16, line 394-396 in the tracked version).

9- Fig 7A, label of the polymerase region in the scheme indicates “RpRp“ instead of “RdRp“.

Now corrected.

Reviewer 2:

This study analyzed host and virus-derived insertions in the hypervariable region (HVR) of the HEV ORF1 polyprotein. While these genomic rearrangements associated with an increased HEV replication have been characterized by different teams the underlying molecular and cellular mechanisms involved in this phenomenon are unknown.

The authors used different constructions based on the introduction of insertions in a luciferase replicon and a full-length viral genome of the Kernow-C1-p6 strain. They also tested the effect of these insertions on ribavirin antiviral activity.

Despite well-conducted experiments, the authors failed to identify a common mechanism involved in the enhanced replication conferred by these insertions and suggest that combinatory factors previously suspected like post translational modifications sites, NLS sequences and HVR flexibility explain the pro-viral effect.

The authors also showed that increased replication due to insertions did not affect the susceptibility to ribavirin.

We would like to thank the reviewer for the positive feedback. We believe that the common mechanism is the combinatory effect of different viral determinants and that our data set is novel in this regard. Please find our response to each comment below. We appreciate the time invested to help improve our manuscript.

Comments :

1. Nucleotide insertions of viral and host origin have been described in the ORF1 HVR of the HEV genome in several studies including Shukla PNAS 2011 ; Shukla J Virol 2012 ; Nguyen J Gen Virol 2012 ; Lhomme J Virol 2014 ; Lhomme Frontiers in Microbiol 2020 ; Papp Scientific reports 2022 ; Biederman Liver international 2022. In this study, 1 viral-derived insertion (major variant, 52.4%) and 2 host-derived insertions (minority variants, 3.2% and 10.7%) were identified from a single chronically infected patient previously treated with ribavirin. Did the authors screened other patients with a similar clinical profile to estimate the frequency of these genetic rearrangements or insertions ?

This is an interesting point. We screened other patients for nucleotide insertion employing our HVR amplification methods followed by Illumina sequencing to avoid laborious clonal sequencing. We noted that the analysis of the NGS-based sequences is a major hurdle for the identification of new insertions due to missing reference information for mapping the short reads to. To overcome this obstacle, we are currently developing a bioinformatic pipeline to readily identify insertions in (chronic) HEV patients (Response letter Figure 2). We believe that this tool will be of huge interest to the community and we hope to present it in the near future.

[redacted]

2. *After genetic engineering of viral and host-derived insertions (TRIM22, SERPINA1) into the Kernow-c1 p6 backbone, is there a relationship between the replicative capacity in vitro of these constructions and the proportions of variants identified in vivo ? This point should be discussed.*

Thank you for this comment. We now added a statement to this question in the discussion section that there was no correlation of the proportion of the different variants with the replication efficiency *in vitro* as both host identified insertion as well as the duplications replicated to similar efficiencies (page 19 line 465-467 in the tracked version).

3. *Only a fragment of host gene is inserted in HVR (for instance a fragment of TRIM22 or SERPINA1). Is there a similar effect on the replicative capacity if another fragment (same length) of the same gene is inserted ?*

This is an interesting question. To answer this question, we generated constructs harboring other fragments of the TRIM22 gene, which included a zinc-finger and a coiled-coil domain fragment. As depicted in our new supplementary Fig. 1, these constructs did not replicate to levels compared to the original TRIM22 insertion constructs, indicating determinants beyond the length of insertions. Please also refer to our answer to question 2 of reviewer 1. These results were now included in the manuscript (page 9 line 216-220 in the tracked version).

4. *Data obtained with the replicon system show that increased replication efficiency did not affect ribavirin susceptibility. It is interesting to note that a similar observation was made with polymerase mutations associated to ribavirin therapy. The mechanism of action of ribavirin on HEV is uncertain and no key mutations associated to ribavirin resistance has been characterized.*

It is certainly an interesting observation and demonstrates no resistance mechanism of the different insertions to ribavirin. One of the proposed modes of action of RBV is a direct mutagenic effect on viral genomes, inducing mismatches and subsequent nucleotide substitutions. These transition events can drive the already error-prone viral replication beyond an error threshold, causing viral population extinction. In contrast, the expanded heterogeneous viral population can facilitate selection of mutant viruses with enhanced replication fitness without causing ribavirin resistance. We added a sentence in the discussion section mentioning that no key mutations and insertions with ribavirin resistance have been identified (page 20, line 470-471 in the tracked version).

5. *Characterization of databank-derived HEV insertions*

Line 220 : several insertions characterized by Lhomme in the HVR including the host insertion ITIH were reported for the first time in J Virol 2014, the others in Frontiers microbiology 2020.

Line 224 : The insertion GATM was detected after replication in a cell culture systems, not in vivo.

Thank you for pointing this out. We corrected the issues accordingly.

6. *A crucial point is to understand the mechanism for the observed replication advantage conferred by viral and host-derived insertions in the HVR. The use of AlphaFold2 to analyze the impact of insertions on HVR could be inappropriate because HVR is an intrinsically disordered region (Purdy PLoS One 2012). Therefore the low prediction confidence score observed for the HVR with or without insertions was expected.*

We fully agree that the region of interest is disordered and has been analyzed with other HEV sequences. However, with the identification of novel insertions, we believe that it is important to investigate these structural properties of each novel viral isolate, which were here identified in vivo.

7. The influence of most viral and host-derived insertions described in the present manuscript on post-translational modifications has been previously studied using different bioinformatic tools. In addition, Kenney and Meng showed that Lysine residues in insertions were important for enhanced virus replication (Kenney J Virol 2015).

We agree that the different HVR insertions have been analyzed before by bioinformatic tools. However, we here included novel insertions as well as novel bioinformatic and structural tools that have not been implemented so far in the literature. In addition, we provide for the first-time functional data to this novel and reported insertion, which in combination with the bioinformatic and structural analysis complete the over analysis.

8. Lines 307-334 : the added value of the alanine scanning of the NLS sequence in the RPS17 insertion of p6 is not clear because these experiments were previously performed (Kenney J Virol 2015). The importance of the protein sequence of the insertions rather than the RNA sequence/structure has been previously reported.

In the study by Kenney et al. 2015, the authors have performed similar experiments in mutating the different lysine residues. However, we aimed to confirm and extend these findings with a more comprehensive set of viral mutants and a more efficient cell culture model. Here, we tested a codon optimized version of the RPS17, mutated K49 with all possible codons and a combination of all different constructs. Furthermore, we have virus infection efficiencies of about 40-60% of infected cells with viral titers over 5×10^5 FFU/ml, whereas in the publication from 2015, infection rates of 4% were shown. The importance of the protein structure of HVR insertion has been shown to the best of our knowledge only for the strain 47832c (Scholz et al. Viruses 2021). We here confirm and extend these findings to another HEV strain with other methodologies. We believe that these independent obtained results are important for the scientific community and HEV field. We now added as statement in the discussion section to mention the findings by Scholz et al. (page 20, 468-488 in the tracked version).

9. The 19 aminoacid positions that are different between p1 and p6 (in addition to the inserted RPS17) could play a role in HEV replication. The authors showed that the mutation A220T in the methyltransferase region and HVR mutations in close proximity to the insertion site could restore a replication similar to p6 with the replicon system (but not virus production with the recombinant full-length genome). Are these observations described for Kernow strain generalizable?

Thank you for that question. As depicted in Fig. 7E, F and G, we tested whether the A220T and RPS17/FR phenotypes could be transferred to another non-Kernow HEV-3 strain (83-2) that does not have any insertions. The threonine at position 220 is already encoded in the 83-2 genome, therefore only the RPS17 insertion and the RPS17/FR insertion could be cloned into the 83-2 HVR. The RPS17 insertion increased HEV replication only minimally, while the 83-2/RPS17/FR genome reached levels close to those of p6. These results show that the made observation can be transferred to another HEV-3 strain.

Reviewer #3:

The manuscript written by Wißing et al. on Genetic determinants of host-and virus- derived insertions for hepatitis E virus replication is very interesting and innovative, providing insights in understanding the adaptation potential of circulating HEV strains and genetic determinants for viral replication. It deserves publication. All bioinformatic analyses and in vitro experiments are well described and performed. However, some minor adjustments should be addressed before.

We would like to thank the reviewer for the overall positive comment and appreciate the time invested to help improve our manuscript.

1) The most abundant type of rearrangement were duplications of HEV sequences (98 clones, 52.4 %), mostly of the HVR into the HVR.

In figure 1 and in the text: for duplication 1 and duplication 2, it would be good to add in the panel D the information of the location of the original sequence that is duplicated (N aa from position X to position Y)

As suggested, we now included in the Fig. 1D the duplications information including the location of the original sequence.

2) In figure 5, regarding the synonymous mutant a (K49K AAG ->AAA) why in panel H the plasmid with mutant a Lysin K does not enter in the nucleus? It should be the K49A mutant? This experiment has been done? Please describe better this part.

In Fig. 5H, the k is the nomenclature that represents the mutant described in Fig. 5B that harbors mutations (KR32/33AA KK44/45AA K49A) and not the single lysine mutation. Sorry for that confusion in the nomenclature. For this mutant, the nuclear translocation signal is fully disrupted and therefore no signal in the nucleus. We now added more information to the nomenclature in the results part to better explain the mutant construct k (page 15, line 365).

REVIEWERS' COMMENTS

Reviewer #1 (Remarks to the Author):

The revised manuscript as well as the point by point reply letter is addressing most points raised by the reviewers. In detail, the change of structure suggested by the reviewer has not been implemented but explanation by the authors are convincing enough to not ask further changes.

Overall, I think that the authors satisfactorily addressed the main points and improved the quality of their study.

Reviewer #2 (Remarks to the Author):

The responses provided by the investigators are satisfactory

Reviewer #3 (Remarks to the Author):

The text has improved a lot thanks to the changes suggested by the revision.

However, some small adjustments should still be made from my previous comments.

1) The most abundant type of rearrangement were duplications of HEV sequences (98 clones, 52.4 %), mostly of the HVR into the HVR. I previously asked to add in figure 1 and in the text the information of the location of the original sequence that is duplicated.

The authors now included in the Fig. 1D the duplications information including the location of the original sequence. But for me this new part of the figure is confusing.

a) I would suggest showing the duplication 2 still below the sequence (as it was in the previous figure, before revision).

b) Ok to have the black boxes referring to duplicated sequence snippets.

c) I don't understand why in duplication 1 all the amino acids are reported, including the non mismatches to the reference strain Kernow-C1-p1 (shown here SQVDA.....RTRRLL) and in duplication 2 no, with all the dots to indicate the same amino acids as reference strain Kernow-C1-p1 (G.....). I would suggest using the same way to show duplication 1 and 2.

If I understand correctly, duplication 1 duplicates the marked HVR region which is different from the reference, while duplication 2 duplicates the HVR region similar to the wt strain Kernow-C1-p1 with the exception of a G?

To my opinion, you have 2 options: either insert all amino acids for duplication 2 as well, regardless if they are mismatches or not, or insert only the mismatches in duplication 1.

2) Regarding figure 5, thanks for the explanation and adding more information at page 15, line 330. However, in the legend of panel H, I would change the lysine mutant k with the mutant construct k.

Response to comments raised by Reviewer #3

The text has improved a lot thanks to the changes suggested by the revision. However, some small adjustments should still be made from my previous comments.

1) The most abundant type of rearrangement were duplications of HEV sequences (98 clones, 52.4 %), mostly of the HVR into the HVR. I previously asked to add in figure 1 and in the text the information of the location of the original sequence that is duplicated. The authors now included in the Fig. 1D the duplications information including the location of the original sequence. But for me this new part of the figure is confusing.

a) I would suggest showing the duplication 2 still below the sequence (as it was in the previous figure, before revision).

We thank the reviewer for the suggestions to further improve our manuscript. We now moved the duplication 2 below the sequence as in previous version.

b) Ok to have the black boxes referring to duplicated sequence snippets.

Thank you for agreeing to our suggestion.

c) I don't understand why in duplication 1 all the amino acids are reported, including the non mismatches to the reference strain Kernow-C1-p1 (shown here SQVDA.....RTRRLL) and in duplication 2 no, with all the dots to indicate the same amino acids as reference strain Kernow-C1-p1 (G.....). I would suggest using the same way to show duplication 1 and 2. If I understand correctly, duplication 1 duplicates the marked HVR region which is different from the reference, while duplication 2 duplicates the HVR region similar to the wt strain Kernow-C1-p1 with the exception of a G?

To my opinion, you have 2 options: either insert all amino acids for duplication 2 as well, regardless if they are mismatches or not, or insert only the mismatches in duplication 1.

Thank you for pointing this out. In the end we probably made it more complicated than we intended to. In the new Fig 1d we now use the same way of visualization for both duplications. Our initial idea was to highlight differences in the two duplications, but we agree, it is easier to understand this way.

2) Regarding figure 5, thanks for the explanation and adding more information at page 15, line 330. However, in the legend of panel H, I would change the lysine mutant k with the mutant construct k.

Thank you for the input. We now changed the label to "mutant construct k".